# Synaptic metaplasticity in binarized neural networks

Axel Laborieux[1✉], Maxence Ernoult[1,2], Tifenn Hirtzlin[1] & Damien Querlioz [1✉]

While deep neural networks have surpassed human performance in multiple situations, they are prone to catastrophic forgetting: upon training a new task, they rapidly forget previously learned ones. Neuroscience studies, based on idealized tasks, suggest that in the brain, synapses overcome this issue by adjusting their plasticity depending on their past history. However, such "metaplastic" behaviors do not transfer directly to mitigate catastrophic forgetting in deep neural networks. In this work, we interpret the hidden weights used by binarized neural networks, a low-precision version of deep neural networks, as metaplastic variables, and modify their training technique to alleviate forgetting. Building on this idea, we propose and demonstrate experimentally, in situations of multitask and stream learning, a training technique that reduces catastrophic forgetting without needing previously presented data, nor formal boundaries between datasets and with performance approaching more mainstream techniques with task boundaries. We support our approach with a theoretical analysis on a tractable task. This work bridges computational neuroscience and deep learning, and presents significant assets for future embedded and neuromorphic systems, especially when using novel nanodevices featuring physics analogous to metaplasticity.

[1] Université Paris-Saclay, CNRS, Centre de Nanosciences et de Nanotechnologies, Palaiseau, France. [2] Unité Mixte de Physique, CNRS, Thales, Université Paris-Saclay, Palaiseau, France. ✉email: axel.laborieux@c2n.upsaclay.fr; damien.querlioz@c2n.upsaclay.fr

In recent years, deep neural networks have experienced incredible developments, outperforming the state-of-the-art, and sometimes human performance, for tasks ranging from image classification to natural language processing[1]. Nonetheless, these models suffer from catastrophic forgetting[2,3] when learning new tasks: synaptic weights optimized during former tasks are not protected against further weight updates and are overwritten, causing the accuracy of the neural network on these former tasks to plummet[4,5] (see Fig. 1a). Balancing between learning new tasks and remembering old ones is sometimes thought of as a trade-off between plasticity and rigidity: synaptic weights need to be modified in order to learn, but also to remain stable in order to remember. This issue is particularly critical in embedded environments, where data are processed in real time without the possibility of storing past data. Given the rate of synaptic modifications, most artificial neural networks were found to have exponentially fast forgetting[6]. This contrasts strongly with the capability of the brain, whose forgetting process is typically described with a power law decay[7], and which can naturally perform continual learning.

The neuroscience literature provides insights about underlying mechanisms in the brain that enable task retention. In particular, it was suggested by Fusi et al.[6,8] that memory storage requires, within each synapse, hidden states with multiple degrees of plasticity. For a given synapse, the higher the value of this hidden state, the less likely this synapse is to change: it is said to be consolidated. These hidden variables could account for activity-dependent mechanisms regulated by intercellular signaling molecules occurring in real synapses[9,10]. The plasticity of the synapse itself being plastic, this behavior is named "metaplasticity." The metaplastic state of a synapse can be viewed as a criterion of importance with respect to the tasks that have been learned throughout and therefore constitutes one possible approach to overcome catastrophic forgetting.

Until now, the models of metaplasticity have been used for idealized situations in neuroscience studies, or for elementary machine learning tasks such as the Cart-Pole problem[11]. However, intriguingly, in the field of deep learning, binarized neural networks[12] (or the closely related XNOR-NETs[13]) have a remote connection with the concept of metaplasticity, also reminiscent, in neuroscience, of the multistate models with binary readout[14]. This connection has never been explored to perform continual learning in multilayer networks. Binarized neural networks are neural networks whose weights and activations are constrained to the values +1 and −1. These networks were developed for performing inference with low computational and memory cost[15–17], and surprisingly, can achieve excellent accuracy on multiple vision[13,18] and signal processing[19] tasks. The training procedure of binarized neural networks involves a real value associated to each synapse, which accumulates the gradients of the loss computed with binary weights. This real value is said to be "hidden," as during inference, we only use its sign to get the binary weight. In this work, we interpret the hidden weight in binarized neural networks as a metaplastic variable that can be leveraged to achieve multitask learning. Based on this insight, we develop a learning strategy using binarized neural networks to alleviate catastrophic forgetting with strong biological-type constraints: previously presented data can not be stored, nor generated, and the loss function is not task-dependent with weight penalties.

An important benefit of our synapse-centric approach is that it does not require a formal separation between datasets, which also allows the possibility to learn a single task in a more continuous fashion. Traditionally, if new data appears, the network needs to relearn incorporating the new data into the old data: otherwise

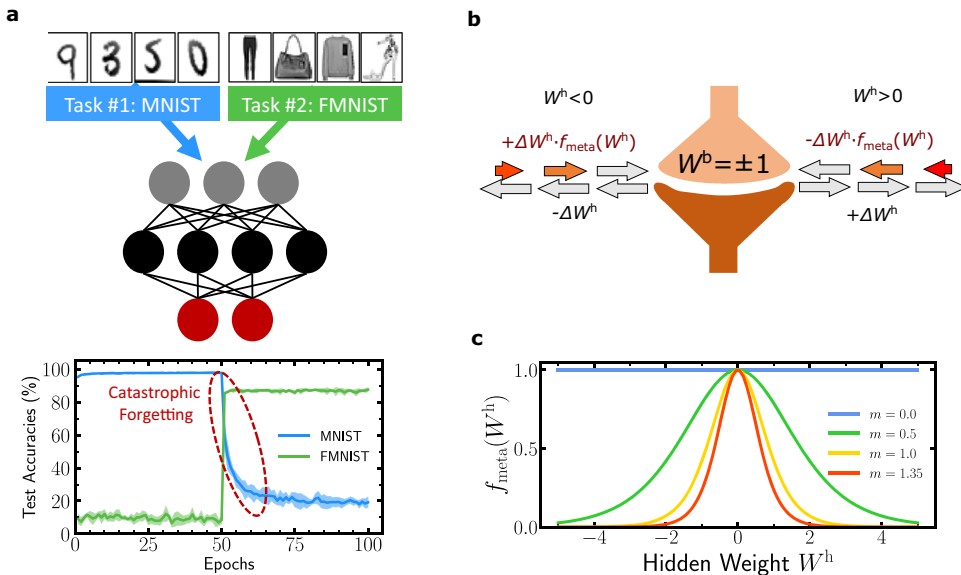

**Fig. 1 Problem setting and illustration of our approach. a** Problem setting: two training sets (here MNIST and Fashion-MNIST) are presented sequentially to a fully connected neural network. When learning MNIST (epochs 0–50), the MNIST test accuracy reaches 97%, while the Fashion-MNIST accuracy stays around 10%. When learning Fashion-MNIST (epochs 50–100), the associated test accuracy reaches 85% while the MNIST test accuracy collapses to ~20% in 25 epochs: this phenomenon is known as "catastrophic forgetting." **b** Illustration of our approach: in a binarized neural network, each synapse incorporates a hidden weight $W^h$ used for learning and a binary weight $W^b = \text{sign}(W^h)$ used for inference. Our method, inspired by neuroscience works in the literature[6], amounts to regarding hidden weights as metaplastic states that can encode memory across tasks and thereby alleviate forgetting. With regards to the conventional training technique of binarized neural network, it consists in modulating some hidden weight updates by a function $f_{\text{meta}}(W^h)$ whose shape is indicated in **c**. This modulation is applied to negative updates of positive hidden weights, and to positive updates of negative hidden weights. $f_{\text{meta}}(|W^h|)$ being a decreasing function, this modulation makes the hidden weight signs less likely to switch back when they grow in absolute value.

the network will just learn the new data and forget what it had already learned. Through the example of the progressive learning of datasets, we show that our metaplastic binarized neural network, by contrast, can continue to learn a task when new data becomes available, without seeing the previously presented data of the dataset. This feature makes our approach particularly attractive for embedded contexts. The spatially and temporally local nature of the consolidation mechanism makes it also highly attractive for hardware implementations, in particular using neuromorphic approaches.

Our approach takes a remarkably different direction than the considerable research in deep learning that is now addressing the question of catastrophic forgetting. Many proposals consist in keeping or retrieving information about the data or the model at previous tasks: using data generation[20], the storing of exemplars[21], or in preserving the initial model response in some components of the network[22]. These strategies do not seem connected to how the brain avoids catastrophic forgetting, need a very formal separation of the tasks, and are not very appropriate for embedded contexts. A solution to solve the trade-off between plasticity and rigidity more connected to ours is to protect synaptic weights from further changes according to their "importance" for the previous task. For example, elastic weight consolidation[3] uses an estimate of the diagonal elements of the Fisher information matrix of the model distribution with respect to its parameters to identify synaptic weights qualifying as important for a given task. Another work[23] uses the sensitivity of the network with respect to small changes in synaptic weights. Finally, in ref. [24], the consolidation strategy consists in computing an importance factor based on path integral. This last approach is the closest to the biological models of metaplasticity, as all computations can be performed at the level of the synapse, and the importance factor is therefore reminiscent of a metaplasticity parameter.

However, in all these techniques, the desired memory effect is enforced by optimizing a loss function with a penalty term, which depends on the previous optimum, and does not emerge from the synaptic behavior itself. This aspect requires a very formal separation of the tasks—the weight values at the end of task training need to be stored—and makes these models still largely incompatible with the constraints of biology and embedded contexts. The highly non-local nature of the consolidation mechanism also makes it difficult to implement in neuromorphic-type hardware.

Specifically, the contributions of the present work are the following:

- We interpret the hidden real value associated to each weight (or hidden weight) in binarized neural networks as a metaplastic variable, we propose a new training algorithm for these networks adapted to learning different tasks sequentially (Alg. 1).
- We show that our algorithm allows a binarized neural network to learn permuted MNIST tasks sequentially with an accuracy equivalent to elastic weight consolidation, but without any change to the loss function or the explicit computation of a task-specific importance factor. More complex sequences such as MNIST- Fashion-MNIST, MNIST- USPS, and CIFAR-10/100 features can also be learned sequentially.
- We show that our algorithm enables to learn the Fashion-MNIST and the CIFAR-10 datasets by learning sequentially each subset of these datasets, which we call the stream-type setting.
- We show that our approach has a mathematical justification in the case of a tractable quadratic binary task where the trajectory of hidden weights can be derived explicitly.

- We adapt our approach to a more complex metaplasticity rule inspired by ref. [8] and show that it can achieve steady-state continual learning. This allows us to discuss the merits and drawbacks of complex and simpler approaches to metaplasticity, especially for hardware implementations of deep learning.

## Results

**Interpreting the hidden weights of binarized neural networks as metaplasticity states**. Synapses in binarized neural networks consist of binary switches that can take either +1 or −1 weights. Learning a task consists in finding a set of binary synaptic values that optimize an objective function related to the task at hand. All synapses share the same plasticity rule and are free to switch back and forth between the two weight values. When learning a second task after the first task, new synaptic transitions between +1 and −1 will overwrite the set of transitions found for the first task, leading to the fast forgetting of previous knowledge (Fig. 1a). This scenario is reminiscent of the neural networks studied in ref. [25] where all synapses are equally plastic and their probability to remain unchanged over a given period of time decreases exponentially for increasing time periods, leading to memory lifetimes scaling logarithmically with the size of the network. Synaptic metaplasticity models were introduced by Fusi et al.[6] to provide long memory lifetimes, by endowing synapses with the ability to adjust their plasticity throughout time—making the plasticity itself plastic. In particular, in this vision, a synapse that is repeatedly potentiated should not increase its weight but rather become more resistant to further depression. In the cascade model[6], plasticity levels are discrete and the probability for a synapse to switch to the opposite strength value decreases exponentially with the depth of the plasticity level. This exponential scaling is introduced to obtain a large range of transition rates, ranging from fast synapses at the top of the cascade where the transition probability is unaffected, to slow synapses that are less likely to switch. Because the metaplastic state only controls the transition probability and not the synaptic strength (i.e., the weight value), it constitutes a "hidden" state as far as synaptic currents are concerned.

The training process of conventional binarized neural networks relies on updating hidden real weights associated with each synapse, using loss gradients computed with binary weights. The binary weights are the signs of the hidden real weights, and are used in the equations of both the forward and backward passes. By contrast, the hidden weights are updated as a result of the learning rule, which therefore affects the binary weights only when the hidden weight changes sign—the detailed training algorithms are presented in Supplementary Algorithms 1 and 2 of Supplementary Note 1. Once the hidden real weight is positive (respectively negative), the binary weight (synaptic strength) is set to +1 (respectively −1), but the synaptic strength will not change if the hidden weight continues to increase toward greater positive (respectively negative) values as a result of the training process. This feature means that hidden weights may be interpreted as analogs to the metaplastic states of the metaplasticity cascade model[6]. However, in conventional binarized neural networks, no mechanism guarantees that when the hidden weight gets updated farther away from zero, the transition to the opposite weight value gets less and less likely. Here, following the insight of ref. [6], we show that introducing such a mechanism yields memory effects.

The mechanism that we propose is illustrated in Fig. 1b, where $W^h$ is the hidden weight and $\Delta W^h$ is the update provided by the learning algorithm, and detailed in Algorithm 1. We introduce a set of functions $f_{meta}$, parameterized by a scalar $m$ and depending

on the hidden weight to modulate the strength of updates in the inverse direction to the sign of the hidden weights. The specific choice of this set of functions is motivated by the conceptual properties that we want our model to share with the cascade model[6]. First, the functions $f_{\mathrm{meta}}$ should be chosen so that the switching strength of the binary weight decreases exponentially with the amplitude of the hidden weight. On the other hand, the switching ability should remain unaffected when the hidden weight is close to zero, making the learning process of such weights analogous to the training of a conventional binarized neural network. We therefore choose a set of functions plotted in Fig. 1c that decrease exponentially to zero as the hidden weight $|W^{\mathrm{h}}|$ approaches infinity, while being flat and equal to one around zero values of $W^{\mathrm{h}}$:

$$f_{\mathrm{meta}}(m, W^{\mathrm{h}}) = 1 - \tanh^2(m \cdot W^{\mathrm{h}}). \quad (1)$$

The parameter $m$ controls the speed at which the decay occurs and constitutes the only hyper-parameter introduced in our approach. More details about the choice of the $f_{\mathrm{meta}}$ function, as well as more implementation details are provided in "Methods." All experiments in this work use adaptive moment estimation (Adam)[26]. Momentum-based training and root-mean-square propagation showed equivalent results. However, pure stochastic gradient descent leads to lower accuracy, as usually observed in binarized neural networks, where momentum is an important element to stabilize training[12,13,17].

**Algorithm 1** Our modification of the BNN training procedure to implement metaplasticity. $\mathbf{W^h}$ is the vector of hidden weights and $W^{\mathrm{h}}$ denotes one component (the same rule is applied for other vectors), $\boldsymbol{\theta}^{\mathrm{BN}}$ are batch-normalization parameters, $\mathbf{U_W}$ and $\mathbf{U_\theta}$ are the parameter updates prescribed by the Adam algorithm[26], $(\mathbf{x}, \mathbf{y})$ is a batch of labeled training data, $m$ is the metaplasticity parameter, and $\eta$ is the learning rate. "·" denotes the element-wise product of two tensors with compatible shapes. The difference between our implementation and the non-metaplastic implementation (recovered for $m = 0$) lies in the condition lines 6–9. $f_{\mathrm{meta}}$ is applied element-wise with respect to $\mathbf{W^h}$. "cache" denotes all the intermediate layers computations needed to be stored for the backward pass. The details of the Forward and Backward functions are provided in Supplementary Note 1.

Input: $\mathbf{W^h}$, $\boldsymbol{\theta}^{\mathrm{BN}}$, $\mathbf{U_W}$, $\mathbf{U_\theta}$, $(\mathbf{x}, \mathbf{y})$, $m$, $\eta$.
Output: $\mathbf{W^h}$, $\boldsymbol{\theta}^{\mathrm{BN}}$, $\mathbf{U_W}$, $\mathbf{U_\theta}$.
1: $\mathbf{W^b} \leftarrow \mathrm{Sign}(\mathbf{W^h})$ ▷Computing binary weights
2: $\hat{\mathbf{y}}$, cache $\leftarrow$ Forward$(\mathbf{x}, \mathbf{W^b}, \boldsymbol{\theta}^{\mathrm{BN}})$ ▷Perform inference
3: $C \leftarrow \mathrm{Cost}(\hat{\mathbf{y}}, \mathbf{y})$ ▷Compute mean loss over the batch
4: $(\partial_{\mathbf{W}}C, \partial_{\boldsymbol{\theta}}C) \leftarrow$ Backward$(C, \hat{\mathbf{y}}, \mathbf{W^b}, \boldsymbol{\theta}^{\mathrm{BN}}, $ cache$)$
   ▷Cost gradients
5: $(\mathbf{U_W}, \mathbf{U_\theta}) \leftarrow \mathrm{Adam}(\partial_{\mathbf{W}}C, \partial_{\boldsymbol{\theta}}C, \mathbf{U_W}, \mathbf{U_\theta})$
6: **for** $W^{\mathrm{h}}$ in $\mathbf{W^h}$ **do**
7: **if** $U_W \cdot W^{\mathrm{b}} > 0$ **then** ▷If $U_W$ prescribes to decrease$|W^{\mathrm{b}}|$
8: $W^{\mathrm{h}} \leftarrow W^{\mathrm{h}} - \eta U_W \cdot f_{\mathrm{meta}}(m, W^{\mathrm{h}})$ ▷Metaplastic update
9: **else**
10: $W^{\mathrm{h}} \leftarrow W^{\mathrm{h}} - \eta U_W$
11: **end if**
12: **end for**
13: $\boldsymbol{\theta}^{\mathrm{BN}} \leftarrow \boldsymbol{\theta}^{\mathrm{BN}} - \eta \mathbf{U_\theta}$
14: **return** $\mathbf{W^h}$, $\boldsymbol{\theta}^{\mathrm{BN}}$, $\mathbf{U_W}$, $\mathbf{U_\theta}$

**Multitask learning with metaplastic binarized neural networks**. We first test the validity of our approach by learning sequentially multiple versions of the MNIST dataset where the pixels have been permuted, which constitutes a canonical benchmark for continual learning[2]. We train a binarized neural network with two hidden layers of 4096 units using Algorithm 1 with several

metaplasticity $m$ values and 40 epochs per task (see "Methods"). Figure 2 shows this process of learning six tasks. The conventional binarized neural network ($m = 0.0$) is subject to catastrophic forgetting: after learning a given task, the test accuracy quickly drops upon learning a new task. Increasing the parameter $m$ gradually prevents the test accuracy on previous tasks from decreasing with eventually the $m = 1.35$ binarized neural network (Fig. 2d) managing to learn all six tasks with test accuracies comparable with the 97.4% test accuracy achieved by the BNN trained on one task only (see Table 1).

Figure 2g, h shows the distribution of the metaplastic hidden weights after learning Task 1 and Task 2 in the second layer. The consolidated weights of the first task correspond to hidden weights between zero and five in magnitude. We observe in Fig. 2g that around $10^7$ of binary weights still have hidden weights near zero after learning one task. These weights correspond to synapses that repeatedly switched between $+1$ and $-1$ binary weights during the training of the first task, and are thus of little importance for the first task. These synapses were therefore not consolidated, and are then available for learning another task. After learning the second task, we can distinguish between hidden weights of synapses consolidated for Task 1 and for Task 2.

Table 1 presents a comparison of the results obtained using our technique with a random consolidation of weights, and with elastic weight consolidation[3], implemented on the same binarized neural network architecture (see "Methods" for the details of EWC adaptation to BNNs). We see that the random consolidation approach does not allow multitask learning. On the other hand, our approach achieves a performance similar to elastic weight consolidation for learning six permuted MNISTs with the given architecture, although unlike elastic weight consolidation, the consolidation does not require changing the loss function and thus does not require task boundaries.

We also perform a control experiment by decreasing the learning rate between each task. The initial learning rate is divided by ten for each new task, as this schedule provided the best results (see Supplementary Note 7). This technique achieves some memory effects but is not as effective as other consolidation methods: uniformly scaling down the learning rate for all synapses at once does not provide a wide range of synaptic plasticity where important synapses are consolidated and less important ones are more plastic.

Figure 3 shows a more detailed analysis of the performance of our approach when learning up to ten MNIST permutations, and for varying sizes of the binarized neural network, highlighting the connection between network size and its capacity in terms of number of tasks. We see that in this harder situation, elastic weight consolidation is more efficient with respect to the network size, especially for smaller networks. Figure 3c, d shows the accuracy obtained when all tasks are learned at once, for a non-metaplastic and metaplastic binarized neural network. This result quantifies the capacity reduction induced by sequential learning. We also compare our approach with "synaptic intelligence" introduced in ref. [24] in Supplementary Fig. 1. This approach features task boundaries as in the case of elastic weight consolidation, but can perform most operations locally, bringing it closer to biology, while retaining near-equivalent accuracy to elastic weight consolidation[24]. Contrary to elastic weight consolidation, synaptic intelligence does not adapt well to a binarized neural network: the importance factor involving a path integral cannot be computed in a natural manner using binarized weights (see Supplementary Note 3), leading to poor performance (Supplementary Fig. 1b). On the other hand, synaptic intelligence applied to full precision neural networks requires less synapses than our binarized approach for equivalent accuracy (Supplementary Fig. 1a), as binarized neural networks always require

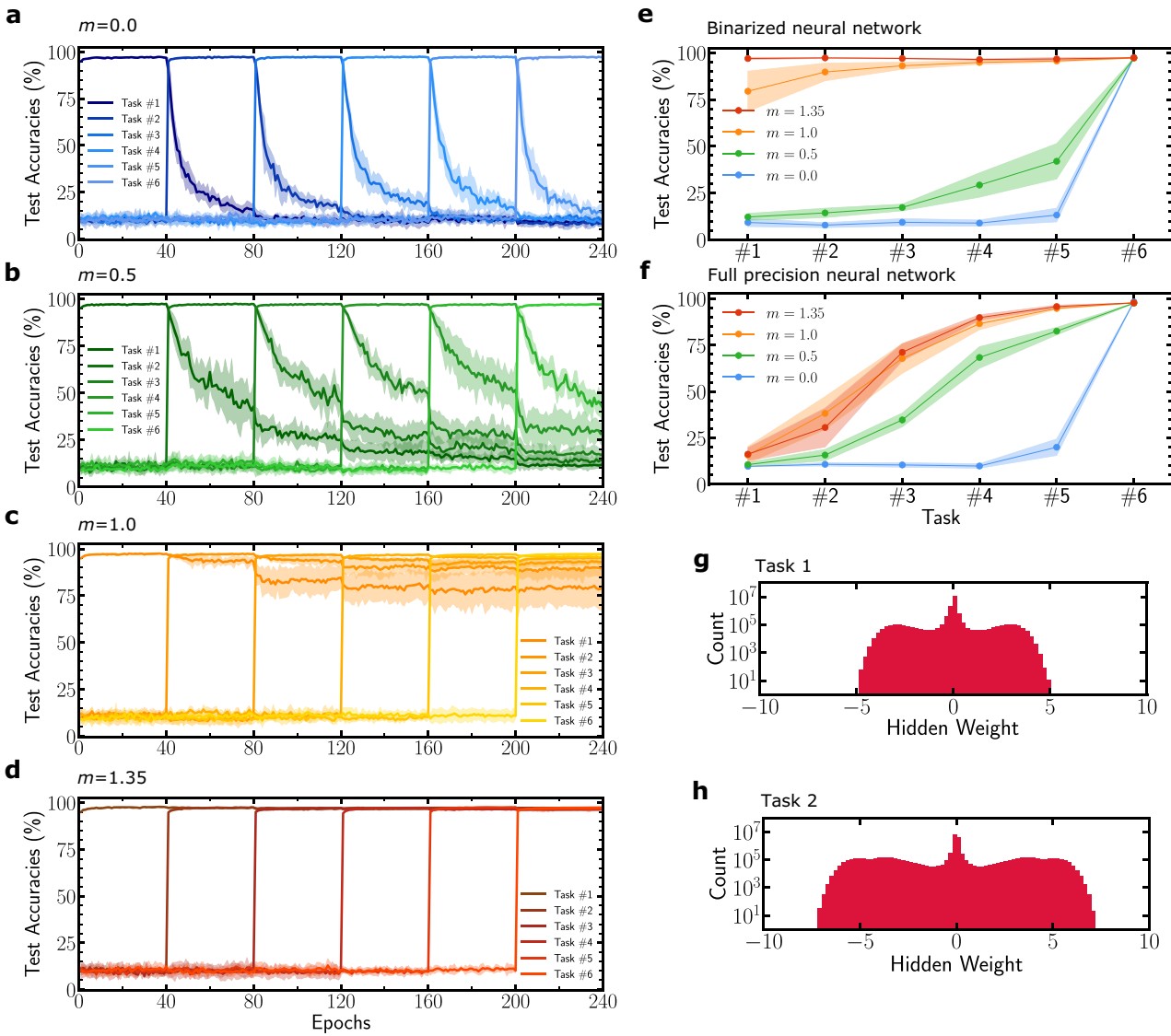

**Fig. 2 Permuted MNIST learning task. a–d** Binarized neural network learning six tasks sequentially for several values of the metaplastic parameter $m$. **a** $m = 0$ corresponds to a conventional binarized neural network **b** $m = 0.5$, **c** $m = 1.0$, **d** $m = 1.35$. Curves are averaged over five runs and shadows correspond to one standard deviation. **e**, **f** Final test accuracy on each task after the last task has been learned. The dots indicate the mean values over five runs, and the shaded zone one standard deviation. **e** Corresponds to a binarized neural network and **f** corresponds to our method applied to a real valued weights deep neural network with the same architecture. **g**, **h** Hidden weights distribution of a $m = 1.35$, two hidden layers of 4096 units binarized neural network after learning for 40 epochs one permuted MNIST (**g**) and two permuted MNISTs (**h**).

**Table 1 Binarized neural network test accuracies on six permuted MNISTs at the end of training for different settings.**

|  | No consolidation ($m = 0.0$) | Random consolidation | Learning rate decay | Elastic weight consolidation | Metaplasticity ($m = 1.35$) |
|---|---|---|---|---|---|
| Task 1 | 9.2 ± 2.2 | 29.0 ± 2.9 | 71.1 ± 6.5 | 96.8 ± 0.7 | 96.9 ± 0.6 |
| Task 2 | 7.8 ± 1.3 | 29.0 ± 4.2 | 87.2 ± 2.6 | 97.2 ± 0.2 | 97.2 ± 0.3 |
| Task 3 | 9.3 ± 2.0 | 32.7 ± 4.7 | 86.1 ± 2.9 | 96.9 ± 0.2 | 96.9 ± 0.2 |
| Task 4 | 9.0 ± 1.7 | 35.1 ± 4.1 | 63.7 ± 5.6 | 96.6 ± 0.2 | 96.4 ± 0.4 |
| Task 5 | 13.2 ± 3.7 | 47.7 ± 8.8 | 75.1 ± 2.5 | 96.8 ± 0.3 | 96.7 ± 0.8 |
| Task 6 | 97.4 ± 0.2 | 96.8 ± 0.2 | 93.9 ± 0.2 | 96.8 ± 0.3 | 97.3 ± 0.1 |

We indicate mean and standard deviation over five trials, for a conventional (non-metaplastic) BNN ($m = 0.0$), a task-dependent learning rate decay scheduler, consolidation of synapses with random importance factors, elastic weight consolidation (EWC)[3] computed with parameter $\lambda_{EWC} = 5 \cdot 10^3$, and our metaplastic binarized neural network approach with parameter $m = 1.35$.

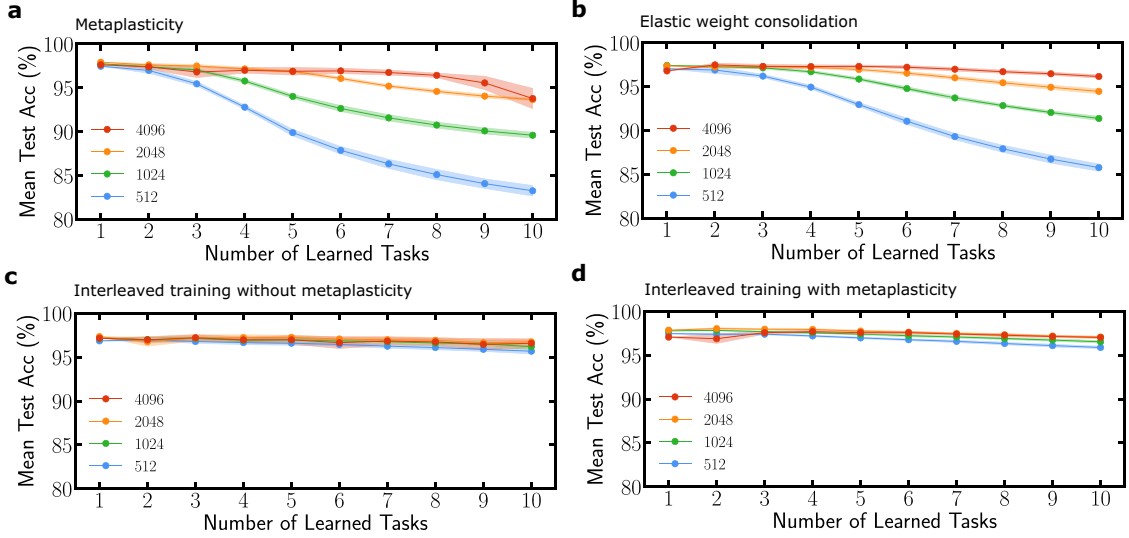

**Fig. 3 Influence of the network size on the number of tasks learned. a, b** Mean test accuracy over tasks learned so far for up to ten tasks. Each task is a permuted version of MNIST learned for 40 epochs. The binarized neural network architecture consists of two hidden layers of a variable number of hidden units ranging from 512 to 4096. **a** Uses metaplasticity with parameter $m = 1.35$ and **b** uses elastic weight consolidation with $\lambda_{EWC} = 5000$. The decrease in mean test accuracy comes from the impossibility to learn new tasks because too many weights are consolidated. Results for non-sequential (interleaved) training for **c** a non-metaplastic and **d** a metaplastic binarized neural network. In this situation, each point is an independent training experiment performed on the corresponding number of tasks. All curves are averaged over five runs and shadow areas denote one standard deviation.

more synapses than full precision ones to reach equivalent accuracy[17,27]. Our technique, therefore, approaches but does not match the accuracy of task-separated approaches. The major motivation of our approach is the possibilities allowed by the absence of task boundaries, such as the stream learning situation investigated in the next section.

Finally, as a control experiment, we also applied Algorithm 1 to a full precision network, except for the weight binarization step described in line one. Figure 2e, f shows the final accuracy of each task at the end of learning for a binarized neural network and a real valued weights deep neural network respectively, with the same architecture. The full precision network final test accuracy of each task for the same range of $m$ values cannot retain more than three tasks with accuracy above 90%. This result highlights that our weight consolidation strategy is tied specifically to the use of hidden weights.

Hidden weights in a binarized neural network and real weights in a full precision neural network respectively possess fundamentally different meanings. In full precision networks, the inference is carried out using the real weights, in particular the loss function is also computed using these weights. Conversely in binarized neural networks, the inference is done with the binary weights and the loss function is also evaluated with these binary weights, which has two major consequences. First, the hidden weights do not undergo the same updates as the weights of a full precision network. Second, a change on a synapse whose hidden weight is positive and which is prescribed a positive update consequently will not affect the loss, nor its gradient at the next learning iteration since the loss only takes into account the sign of the hidden weights. Hidden weights in binarized neural networks consequently have a natural tendency to spread over time (Fig. 2g, h), and they are not weights properly speaking. Supplementary Fig. 3 illustrates this difference visually. In a full precision neural network, "important" weights for a task converge to an optimum value minimizing the loss. By contrast, in a binarized neural network, when a binarized weight has stabilized to its optimum value, its hidden weight keeps increasing, thereby clearly indicating that the synapse should be consolidated. At the end

of the paper, we provide a deeper mathematical interpretation of this intuition.

We also tested the capability of our binarized neural network to learn sequentially different datasets, in several situations. We first investigated the sequential training of the MNIST and the Fashion-MNIST dataset, presenting apparel items[28]. While a non-metaplastic network rapidly forgets the first dataset when the second one is trained (Fig. 4a), an optimized metaplastic network learns both tasks with accuracies near the ones achieved when the tasks are learned independently (Fig. 4b). More details and more results are presented in Supplementary Note 9. Figure 4c presents the sequential training of two closely related datasets: MNIST, and of a second handwritten digits dataset (United States Postal Services). A small amount of data is used in this experiment to keep the balance between the two datasets (see Supplementary Note 10). The baselines are non-metaplastic networks obtained by partitioning the metaplastic network into two equal parts (each featuring half the number of hidden neurons), and trained independently on each task. We see that the metaplastic network learns sequentially both datasets successfully with accuracies above the baselines, suggesting that for a fixed number of hidden neurons, metaplasticity can provide an increase in capacity. Figure 4d presents a variation of this situation with the same baselines, and where the metaplastic network is this time designed with a number of parameters doubled with regards to the baselines (see Supplementary Note 10). In that case, the accuracy of the sequentially trained metaplastic network still succeeds at matching, but does not exceed the non-sequentially trained baselines. Finally, we investigated a situation of class incremental learning of the CIFAR-10 (Fig. 4e, f) and CIFAR-100 (Fig. 4g, h) datasets. We use a convolutional neural network with convolutional layers pretrained on ImageNet, and a metaplastic classifier (see Supplementary Note 11). The classes of these datasets are divided into two subsets and trained sequentially. While in the non-metaplastic network (Fig. 4e–g), the first subset of classes is forgotten rapidly when the second is trained, in the metaplastic one (Fig. 4f–h), good accuracy is achieved, which remains below the one obtained with non-sequentially trained classes. Better

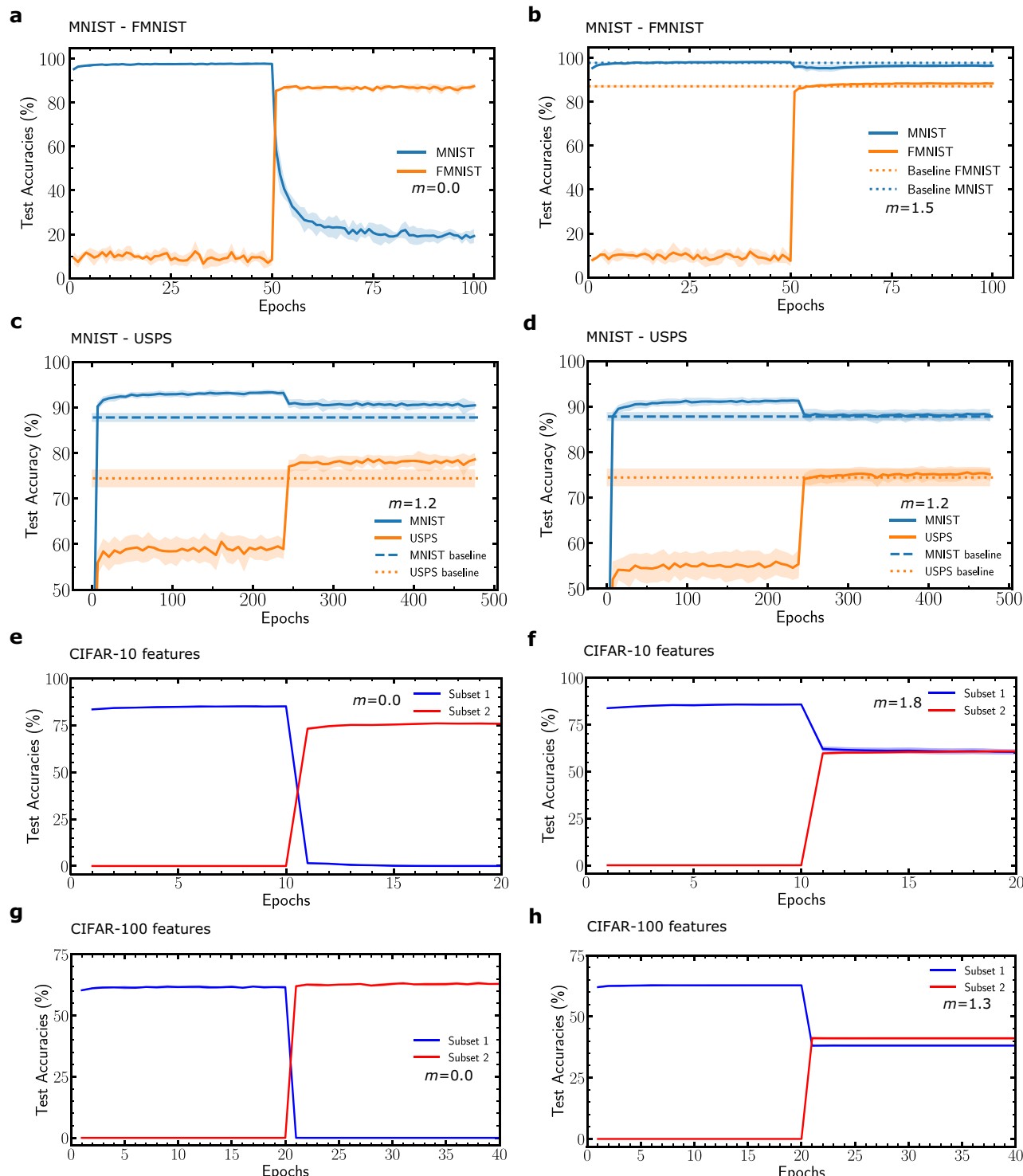

**Fig. 4 Sequential learning on various datasets.** Binarized neural network learning MNIST and Fashion-MNIST sequentially **a** without metaplasticity and **b** with metaplasticity. **c** Sequential training of the MNIST and USPS datasets of handwritten digits. The baselines correspond to the accuracy reached by non-metaplastic networks with half the number of neurons trained independently on each task. **d** Presents the same experiment as **c**, with a metaplastic network featuring a doubled number of parameters with regards to the baselines. **e**, **f** Test accuracy when learning sequentially two subsets of CIFAR-10 classes from features extracted by a pretrained ResNet on ImageNet (see Supplementary Note 11). **g**, **h** Same experiment with CIFAR-100 features. All curves except **c** and **d** are averaged over five runs. **c** and **d** are averaged over fifty runs due to the small amount of data (see Supplementary Note 10). Shadows correspond to one standard deviation.

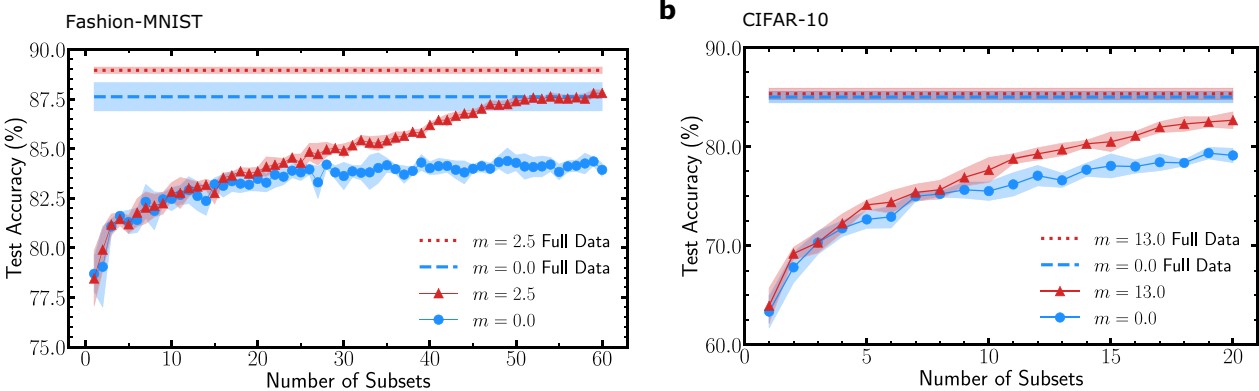

**Fig. 5 Stream learning experiments. a** Progressive learning of the Fashion-MNIST dataset. The dataset is split into 60 parts consisting of only 1000 examples, and containing all ten classes. Each sub dataset is learned for 20 epochs. The dashed lines represent the accuracies reached when the training is done on the full dataset for 20 epochs so that all curves are obtained with the same number of optimization steps. **b** Progressive learning of the CIFAR-10 dataset. The dataset is split into 20 parts, consisting of only 2500 examples. Each sub dataset is learned for 200 epochs. The dashed lines represent the accuracies reached when the training is done on the full dataset for 200 epochs. Shadows correspond to one standard deviation around the mean over five runs.

performance can be achieved if we allow the neurons to have independent thresholds for the two subsets (see Supplementary Note 11).

**Stream learning: learning one task from subsets of data.** We have shown that the hidden weights of binarized neural networks can be used as importance factors for synaptic consolidation. Therefore, in our approach, it is not required to compute an explicit importance factor for each synaptic weight: our consolidation strategy is carried out simultaneously with the weight update as consolidation only involves the hidden weights. The absence of formal dataset boundaries in our approach is important to tackle another aspect of catastrophic forgetting where all the training data of a given task is not available at the same time. In this section, we use our method to address this situation, which we call "stream learning": the network learns one task but can only access one subset of the full dataset at a given time. Subsets of the full dataset are learned sequentially and the data of previous subsets cannot be accessed in the future.

We first consider the Fashion-MNIST dataset, split into 60 subsets presented sequentially during training (see "Methods"). The learning curves for regular and metaplastic binarized neural networks are shown in Fig. 5a, the dashed lines corresponding to the accuracy reached by the same architecture trained on the full dataset after full convergence. We observe that the metaplastic binarized neural network trained sequentially on subsets of data performs as well as the non-metaplastic binarized neural network trained on the full dataset. The difference in accuracy between the baselines can be explained by our consolidation strategy gradually reducing the number of weights able to switch, therefore acting as a learning rate decay (the mean accuracy achieved by a binarized neural network with $m = 0$ trained with a learning rate decay on all the data is 88.8%, equivalent to the metaplastic baseline in Fig. 5a).

In order to see if the advantage provided by metaplastic synapses holds for convolutional networks and harder tasks, we then consider the CIFAR-10 dataset, with a binarized version of a Visual Geometry Group (VGG) convolutional neural network (see "Methods"). CIFAR-10 is split into 20 sub datasets of 2500 examples. The test accuracy curve of the metaplastic binarized neural network exhibits a gap with baseline accuracies smaller than the non-metaplastic one. Our metaplastic binarized neural network can thus gain new knowledge from new data without

forgetting previously learned unavailable data. Because our consolidation strategy does not involve changing the loss function and the batch-normalization settings are common across all subsets of data, the metaplastic binarized neural network gains new knowledge with each subset of data without any information about subsets boundaries. This feature is especially useful for embedded applications, and is not currently possible in alternative approaches of the literature to address catastrophic forgetting.

**Mathematical interpretation.** We now provide a mathematical interpretation for the hidden weights of binarized neural networks, also illustrated graphically in Supplementary Note 6. We show in archetypal situations that the larger a hidden weight gets while learning a given task, the bigger the loss increase upon flipping the sign of the associated binary weight, and consequently the more important they are with respect to this task. For this purpose, we define a quadratic binary task, an analytically tractable and convex counterpart of a binarized neural network optimization task. This task, defined formally in Supplementary Note 5, consists in finding the global optimum on a landscape featuring a uniform (Hessian) curvature. The gradient used for the optimization is evaluated using only the signs of the parameters $\mathbf{W}^{\mathrm{h}}$ (Fig. 6a), in the same way that binarized neural networks employ only the sign of hidden weights for computing gradients during training. In Supplementary Note 5, we demonstrate theoretically that throughout optimization on the quadratic binary task, if the uniform norm of the weight optimum vector is greater than one, the hidden weights vector diverges. Figure 6a shows an example in two dimensions where such a divergence is seen. This situation is reminiscent of the training of binarized neural networks on practical tasks, where the divergence of some hidden weights is observed. In the particular case of a diagonal Hessian curvature, a correspondence exists between diverging hidden weights and components of the weight optimum greater than one in absolute value. We can derive an explicit form for the asymptotic evolution of the diverging hidden weights while optimizing: the hidden weights diverge linearly: $W_{i,t}^{\mathrm{h}} \sim \widetilde{W}_i^{\mathrm{h}} t$ with a speed proportional to the curvature and the absolute magnitude of the global optimum (see Supplementary Note 5). Given this result, we can prove the following theorem (see Supplementary Note 5):

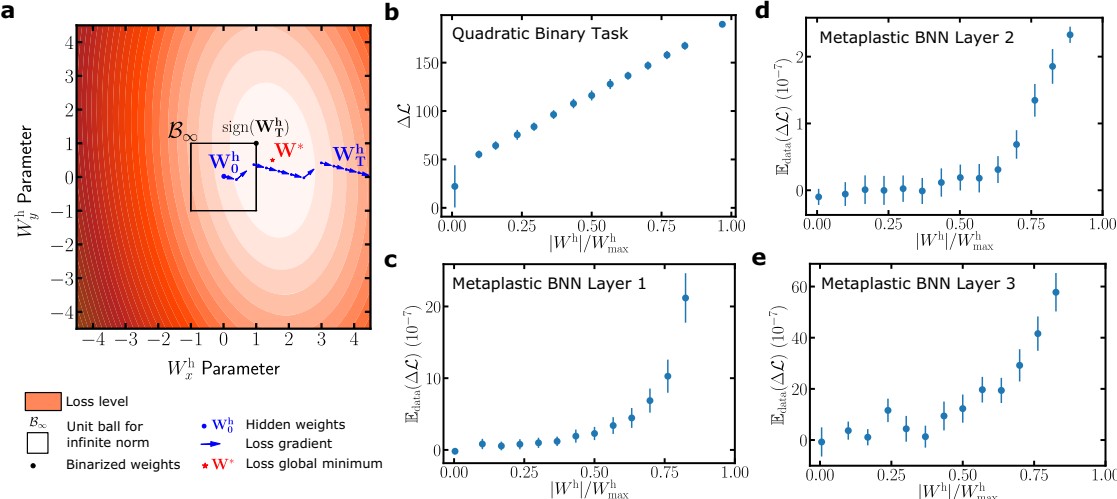

**Fig. 6 Interpretation of the meaning of hidden weights. a** Example of hidden weights trajectory in a two-dimensional quadratic binary task. One hidden weight $W_x^h$ diverges because the optimal hidden weight vector $\mathbf{W}^*$ has uniform norm greater than one (Lemma 2 of Supplementary Note 5). **b** Mean increase in the loss occurred by switching the sign of a hidden weight as a function of the normalized value of the hidden weight, for a 500-dimensional quadratic binary task. The mean is taken by assigning hidden weights to bins of increasing absolute value and the error bars denote one standard deviation around the mean. The leftmost point corresponds to hidden weights staying bounded. **c–e** Increase in the loss occurred by switching the sign of hidden weights as a function of the normalized absolute value of the hidden weight in a binarized neural network trained on MNIST. Each dot is the mean increase over 100 realizations of weights to be switched and the error bars denote one standard deviation. The scales differ because the layers have different numbers of weights and thus different relative importance. See "Methods" for implementation details.

**Theorem 1** *Let* **W** *optimize the quadratic binary task with optimum weight* **W**\* *and curvature matrix* **H**, *using the optimization scheme*: $\mathbf{W}_{t+1}^h = \mathbf{W}_t^h - \eta\mathbf{H} \cdot (\text{sign}(\mathbf{W}_t^h) - \mathbf{W}^*)$. *We assume* **H** *equal to* $\text{diag}(\lambda_1,\dots\lambda_d)$ *with* $\lambda_i > 0, \forall i \in [\![1, d]\!]$. *Then, if* $|W_i^*| > 1$, *the variation of loss resulting from flipping the sign of* $W_{i,t}^b$ *is*:

$$\Delta_i\mathcal{L}(\mathbf{W}_t) \sim 2\lambda_i + 2\frac{|\widetilde{W}_i^h|}{\eta} \quad \text{as} \quad t \to +\infty. \tag{2}$$

This theorem states that the increase in the loss induced by flipping the sign of a diverging hidden weight is asymptotically proportional to the sum of the curvature and a term proportional to the hidden weight. Hence the correlation between high valued hidden weights and important binary weights.

Interestingly, this interpretation, established rigorously in the case of a diagonal Hessian curvature, may generalize to non-diagonal Hessian cases. Figure 6, for example, illustrates the correspondence between hidden weights and high impact on the loss by sign change on a quadratic binary task (Fig. 6b) with a 500-dimensional non-diagonal Hessian matrix (see "Methods" for the generation procedure). Figure 6c–e finally shows that this correspondence extends to a practical binarized neural network situation, trained on MNIST. In this case, the cost variation $\mathbb{E}_{\text{data}}(\Delta\mathcal{L})$ upon switching binary weights signs increases monotonically with the magnitudes of the hidden weights (see "Methods" for implementation details). These results provide an interpretation as to why hidden weights can be thought of as local importance factors useful for continual learning applications.

## Discussion

Addressing catastrophic forgetting with ideas from both neuroscience and machine learning has led us to find an artificial neural network with richer synapses behaviors that can perform continual learning without requiring an overhead computation of task-related importance factors. The continual learning capability of metaplastic binarized neural networks emerges from its intrinsic design, which is in stark contrast with other consolidation strategies[3,23,24]. The resulting model is more autonomous because the optimized loss function is the same across all tasks. Metaplastic synapses enable binarized neural networks to learn several tasks sequentially similarly to related works, but more importantly, our approach takes the first steps beyond a more fundamental limitation of deep learning, namely the need for a full dataset to learn a given task. A single autonomous model able to learn a task from small amounts of data while still gaining knowledge, approaching to some extent the way the brain acquires new information, paves the way for widespread use of embedded hardware for which it is impossible to store large datasets. Other methods have been introduced to train binarized neural networks such as refs. [29] or [30] and provide valuable insights to understand the specificity of binarized networks with respect to continual learning. Helwegen et al.[29] interpret the hidden weight as inertia, which is coherent with the fact that high inertia might correspond to important weights, while Meng et al.[30] link the hidden weight to the natural parameter of a probability distribution over binarized weights, which can be used as a relevant prior to perform continual learning.

A distinctive aspect of continual learning approaches is their behavior when the neural network reaches its capacity in terms of number of tasks. The behavior in the case of our approach can be anticipated from the mathematical interpretation in the previous section: when all hidden weights have started to diverge, i.e., are consolidated for a given task, no weights should be able to learn new tasks. The consequence of this situation is well seen in Supplementary Fig. 4b: when learning ten permuted MNIST tasks, the last task has reduced accuracy, while the first trained tasks retain their original accuracy. This behavior fits well with a large section of the literature on continual learning, multitask learning, where the goal is to learn a given number of tasks[31]. Supplementary Fig. 2 also highlights the relative definitive nature of synaptic consolidation in our approach. We implemented a variation, where the metaplasticity function reaches a hard zero

after a given threshold. We see that the performance on the ten permuted MNIST tasks is only modestly reduced by this change.

This behavior also differentiates our approach from the brain, where a more natural behavior for most networks would be to forget the earliest trained tasks, and replace them with the newly trained ones. In recent years, the literature about metaplasticity has aimed at reproducing this behavior, i.e., a type of "steady-state" continual learning[8,11]. This recent literature can therefore provide leads to implement such behavior in our network. In particular, Benna et al. proposed a metaplasticity model where synapses feature a network of different elements, which all evolve at different time scales[8]. This model can feature a sophisticated memory effect, and one work successfully used this type of synapses in the context of an elementary continual reinforcement learning task related to the Cart-Pole problem[11].

We found that directly applying the metaplasticity rule of ref. [8] in our context does not yield proper memory effects. The explanation stems from the specificity of deep networks: in ref. [8], synaptic updates occur following randomly presented patterns, in an independent and identically distributed fashion. In our continual learning situation, sequential synaptic updates are highly correlated. However, the rule of ref. [8] can still be used as an inspiration to allow steady-state continual learning in our approach. In Supplementary Note 8 and the associated Supplementary Figs. 4 and 5, we provide a learning rule where synapses also feature a network of elements evolving at different time scale adapted for the training of binarized neural networks, leading to a natural forgetting of tasks trained a long time ago when new tasks are trained. Our adaptation consists in modulating the flow between hidden variables, an idea suggested as a perspective in ref. [11] as a way to bridge the gap between conventional continual learning methods and neuroscience-based approaches. We can see in Supplementary Fig. 4c that in this case, when training ten permuted MNIST tasks, the last trained task features the highest accuracy, while the accuracy of the first trained tasks starts to decrease.

This discussion highlights an interplay between the level of continual learning feature and of synaptic complexity. Highly complicated synapses, featuring many equations and hyperparameters, as the ones of refs. [8,11] or the one that we just introduced, can achieve advanced continual learning behaviors. For an artificial system, the richness of highly complex synapses needs to be counterbalanced with their implementation cost. Biology might have experienced a similar dilemma. Evolution seems to have favored synapses exhibiting highly complex metaplastic behaviors[10], although simpler synapses might have been more efficient to implement, suggesting the high computational benefits of complex synapses.

This discussion is natural for software implementations of metaplasticity, and also exists for hardware. In particular, the fact that metaplastic approaches build on synapses with rich behavior resonates with the progress of nanotechnologies, which can provide compact and energy-efficient electronic devices able to mimic neuroscience-inspired models, employing "memristive" technologies[32–35]. Many works in nanotechnologies have shown that a single nanometer-scale device can provide metaplastic behavior[36–40]. The metaplasticity features of these nanodevices vary greatly depending on their underlying physics and technology, but their complexity is analogous to our proposal here. Typically, metaplasticity occurs by transforming the shape of a conductive filament in a continuous fashion. These changes make the device harder to program, and therefore provide a feature that can be analogous to our continuous metaplasticity function $f_{\text{meta}}$. On the other hand, the complicated version of Supplementary Note 8 would be highly challenging to implement with a single nanodevice, based on the current state of nanotechnologies, as

these metaplasticity models require many different states with different time dynamics. Our proposal, as other proposals of complex synapses with multiple variables[41] or stochastic behaviors[42], could therefore be an outstanding candidate for nanotechnological implementations, as it provides rich features at the network level, while remaining compatible with the constraints of technology.

In addition, taking inspiration from the metaplastic behavior of actual synapses of the brain resulted in a strategy where the consolidation is local in space and time. This makes this approach particularly suited for artificial intelligence-dedicated hardware and neuromorphic computing approaches, which can save considerable energy by employing circuit architectures optimized for the topology of neural network models, and therefore limiting data movements[43]. The fact that our metaplasticity approach is entirely local should be put into perspective into the non-local aspects of the overall learning algorithms. First, all our simulations use batch normalization, as it is known to efficiently stabilize the training of binarized neural networks[12,13]. Batch normalization is not, however, a fundamental element of the scheme. Normalization technique that do not involve batches, such as instance normalization[44], layer normalization[45], or online normalization[46] provide more hardware-friendly alternatives. More profoundly, error backpropagation itself is of course nonlocal. Currently, multiple efforts aim at developing more local alternatives to backpropagation[47–49], or at relying on directly bioinspired learning rules[50,51]. We have seen that alternative approaches of the literature to overcome catastrophic forgetting typically rely on the use of additional terms in the loss, are therefore strongly tied to the use of error backpropagation. On the other hand, as our metaplasticity approach is entirely synaptic-centric, it is largely agnostic to the learning rule, and should be adaptable to all these emerging learning approaches. This discussion also evidences the benefit of taking inspiration from biology with regards to purely mathematically motivated approaches: they tend to be naturally compatible with the constraints of hardware developments and can be amenable for the development of energy-efficient artificial intelligence.

In conclusion, we have shown that the hidden weights involved in the training of binarized neural networks are excellent candidates as metaplastic variables that can be efficiently leveraged for continual learning. We have implemented long-term memory into binarized neural networks by modifying the hidden weight update of synapses. Our work highlights that binarized neural networks can be more than a low-precision version of deep neural networks, as well as the potential benefits of the synergy between neurosciences and machine learning research, which for instance aims to convey long-term memory to artificial neural networks. We have also mathematically justified our technique in a tractable quadratic binary problem. Our method allows for online synaptic consolidation directly from model behavior, which is important for neuromorphic dedicated hardware, and is also useful for a variety of settings subject to catastrophic forgetting.

## Methods

**Metaplasticity-inspired training of binarized neural networks**. The binarized neural networks studied in this work are designed and trained following the principles introduced in ref. [12]—specific implementation details are provided in Supplementary Note 2. These networks consist of binarized layers where both weight values and neuron activations assume binary values meaning {+1, −1}. Binarized neural networks can achieve high accuracy on vision tasks[13,18], provided that the number of neurons is increased with regards to real neural networks. Binarized neural networks are especially promising for AI hardware because unlike conventional deep networks, which rely on costly matrix-vector multiplications, these operations for binarized neural networks can be done in hardware with XNOR logic gates and pop-count operations, reducing the power consumption by several orders of magnitude[17].

In this work, we propose an adaptation of the conventional binarized neural network training technique to provide binarized neural networks with metaplastic synapses. We introduce the function $f_{\text{meta}}: \mathbb{R}^+ \times \mathbb{R} \rightarrow \mathbb{R}$ to provide an asymmetry, at equivalent gradient value and for a given weight, between updates toward zero hidden value and away from zero. Algorithm 1 describes our optimization update rule and the unmodified version of the update rule is recovered when $m = 0.0$ due to condition (3) satisfied by $f_{\text{meta}}$. $f_{\text{meta}}$ is defined such that:

$$\forall x \in \mathbb{R}, f_{\text{meta}}(0, x) = 1, \tag{3}$$

$$\forall m \in \mathbb{R}^+, f_{\text{meta}}(m, 0) = 1, \tag{4}$$

$$\forall m \in \mathbb{R}^+, \partial_x f_{\text{meta}}(m, 0) = 0, \tag{5}$$

$$\forall m \in \mathbb{R}^+, \lim_{|x| \to +\infty} f_{\text{meta}}(m, x) = 0. \tag{6}$$

Conditions (4) and (5) ensure that near-zero real values, the weights are free to switch in order to learn. Condition (6) ensures that the farther from zero a real value is, the more difficult it is to make the corresponding weight switch back. In all the experiments of this paper, we use:

$$f_{\text{meta}}(m, x) = 1 - \tanh^2(m \cdot x). \tag{7}$$

The parameter $m$ controls how fast binary weights are consolidated (Fig. 1c). The specific choice of $f_{\text{meta}}$ is made to have a variety of plasticity over large ranges of time steps (iteration steps) with an exponential dependence as in ref. [6]. Specific values of the hyperparameters can be found in Supplementary Note 2.

**Multitask training experiments**. A permuted version of the MNIST dataset consists of a fixed spatial permutation of pixels applied to each example of the dataset. We also train a full precision (32-bits floating point) version of our network with the same architecture for comparison, but with tanh activation function instead of sign. The learned parameters in batch normalization are not binary and therefore cannot be consolidated by our metaplastic strategy. Therefore, in our experiments, the binarized and full precision neural networks have task-specific batch-normalization parameters in order to isolate the effect of weight consolidation on previous tasks test accuracies.

For the control, elastic weight consolidation is applied to binarized neural networks by consolidating the binary weights (and not the hidden weights as the response of the network is determined by the binary weights): both the surrogate loss term, and the Fisher information estimates are computed using the binary weight values. The EWC regularization strength parameter is $\lambda_{\text{EWC}} = 5 \cdot 10^3$. The random consolidation presented in Table 1 consists in computing the same importance factors as elastic weight consolidation but then randomly shuffling the importance factors of the synapses.

**Stream learning experiments**. For Fashion-MNIST experiments, we use a metaplastic binarized neural network of two 1024 units hidden layers. The dataset is split into 60 subsets of 1000 examples each, and each subset is learned for 20 epochs. (All classes are represented in each subset.)

For CIFAR-10 experiments, we use a binary version of VGG-7 similarly to ref. [12], with six convolution layers of 128-128-256-256-512-512 filters and kernel sizes of 3. Dropout with probability 0.5 is used in the last two fully connected layers of 2048 units. Data augmentation is used within each subset with random crop and random rotation.

**Sign switch in a binarized neural network**. Two major differences between the quadratic binary task and the binarized neural network are the dependence on the training data and the relative contribution of each parameter, which is lower in the case of the BNN than in the quadratic binary task. The procedure for generating Fig. 6c–e has to be adapted accordingly. Bins of increasing normalised hidden weights are created, but instead of computing the cost variation for a single sign switch, a fixed amount of weights are switched within each bin so as to increase the contribution of the sign switch on the cost variation. The resulting cost variation is then normalised with respect to the number of switched weights. An average is done over several realizations of the hidden weights to be switched. Given the different sizes of the three layers, the amounts of switched weights per bins for each layer are respectively 1000, 2000, and 100.

**Positive symmetric definite matrix generation**. To generate random positive symmetric definite matrices, we first generate the diagonal matrix of eigen values $\mathbf{D} = \text{diag}(\lambda_1, ..., \lambda_d)$ with a uniform or normal distribution of mean $\mu$ and variance $\sigma$ and ensure that all eigen values are positive. We then use the subgroup algorithm described in ref. [52] to generate a random rotation $\mathbf{R}$ in dimension $d$. This is done by first generating a random rotation $\mathbf{R}_2$ in 2D and iteratively increasing the dimension by sampling a random unitary vector $\mathbf{v}$, then computing $\mathbf{x} = (\mathbf{e}_1 - \mathbf{v}) / \|\mathbf{e}_1 - \mathbf{v}\|$ with $\mathbf{e}_1 = (1, 0, ..., 0)^T$, and finally computing $\mathbf{R}_{n+1} = (\mathbf{I} - 2\mathbf{x}^T \mathbf{x}) \cdot \hat{\mathbf{R}}_n$,

where $\hat{\mathbf{R}}_n$ is a $n + 1 \times n + 1$ matrix where $\hat{\mathbf{R}}_{n,0,0} = 1$, $\hat{\mathbf{R}}_{n,1:,1:} = \mathbf{R}_n$, and $\hat{\mathbf{R}}_{n,0,j} = \hat{\mathbf{R}}_{n,i,0} = 0$. We then compute $\mathbf{H} = \mathbf{R}^T \cdot \mathbf{D} \cdot \mathbf{R}$.

**Reporting summary**. Further information on research design is available in the Nature Research Reporting Summary linked to this article.

## Data availability
All used datasets (MNIST[53], USPS[54], Fashion-MNIST[28], CIFAR-10 and CIFAR-100[55]) are available in the public domain and were obtained from https://pytorch.org/vision/stable/datasets.html for this work.

## Code availability
Throughout this work, all simulations and data analysis are performed using Pytorch 1.1.0. The source codes used in this work are freely available online in the Github repository[56]: https://github.com/Laborieux-Axel/SynapticMetaplasticityBNN.

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

## Acknowledgements

This work was supported by European Research Council Starting Grant NANOINFER (reference: 715872). The authors would like to thank L. Herrera-Diez, T. Dalgaty, J. Thiele, G. Hocquet, P. Bessière, and J. Grollier for discussion and invaluable feedback on the manuscript.

## Author contributions

A.L. developed the PyTorch code used in this project and performed all subsequent simulations. A.L. and M.E. carried the mathematical analysis of the "Mathematical Interpretation." T.H. provided the initial idea for the project, and an initial Numpy version of the code. M.E. and T.H. contributed equally to the project. D.Q. directed the work. All authors participated in data analysis, discussed the results, and co-edited the manuscript.

## Competing interests
The authors declare no competing interests.
