## [Peer Review File · Nature Communications]

Reviewer #1 (Remarks to the Author):

This paper proposes a new method to avoid catastrophic forgetting in binary neural networks (BNN). The main contribution is to first interpret the continuous weights in BNNs as a "metaplastic variable" that indicates the weights importance for remembering the past knowledge. This interpretation is then used to propose a training method which involves modification of Adam update using a function f_meta (line 6 and 7 in algorithm 1). An advantage of the method is that it does not require explicit task boundaries or modification of loss function. Experiments are shown on MNIST-based data (with only one experiment on CIFAR). Comparison with EWC and Random consolidation method are shown. It is not clear if the method really beats the baselines. One mathematical interpretation on a quadratic loss is also discussed.

My overall impression

(1) The method seems useful but the choices made in the algorithm are not justified well and lack a principled approach to attack the problem of continual learning.

(2) Experiments, even though incomplete, do highlight that the method could have an advantage, but the experiments are presented in a misleading way and this needs to be fixed.

Details

====

Regarding point 1:

====

In Line 6 UW is used to make decision to perform metaplastic update. In Line 7 the function f_meta is used to do the update. These choices are not justified. What if we use RMSprop/SGD? Why specifically UW? What if we use some other thresholding mechanism? What about other functions for f_meta ? What kind of properties we need for this function? And why?

When I think about all this, I feel that the jump from neuroscience motivation to an algorithm is too quick. The paper does not have principled ways of connecting the two. This is a big issue.

Regarding Point 2:

====

Looking at the left column in Figure 2, the results seems very good. As we increase m , the forgetting is avoided. Great! But looking into the other results I found this results to be misleading. In Table 1, Metaplasticity is in bold, but it doesn't beat EWC. In fact, in Supplementary Figure 1, EWC is consistently better on a variety of architectures. There the performance is reported for all 10 tasks, while in Table 1 it is only reported for 6 tasks.

My conclusion is that the results are presented in a misleading way and this could have been avoided easily.

Some other fixes would be

- including more datasets and show consistent performance boost. MNIST based tasks are usually not that hard, and it is better to do this on CIFAR-10, Omniglot etc.
- include better baselines (EWC is one but other methods can also be applied).

Two other points that I found problematic

=====

- Why does this not work for full precision? The explanation in line 135, says that the weights contain "a trace of the history of the network updates that is relevant for memory effect". The weights in the full precision also contain the history, so then the principle should also (somehow) transfer to "full precision".

- The abstract says "we highlight a connection between metaplasticity ... and BNN", then the main text changes this to "we interpret the hidden weights ... as metaplastic variable". The former is misleading. The paper's method is motivated from the working mechanism of brain, and in the weights in BNN can then just be interpreted this way.

Reviewer #2 (Remarks to the Author):

This paper approaches the problem of catastrophic forgetting in deep, binary neural networks (BNN) using a version of the BNN training algorithm modified to introduce a form of synaptic metaplasticity. The article is well written and shows some interesting empirical results in support of the usefulness of metaplasticity in continual learning.

I have some comments/concerns:

The authors stress the importance of power-law (rather than exponential) forgetting in the introduction. Was there any evidence of power-law forgetting in their network?

The synaptic metaplasticity model used in this paper is rather ad hoc and heuristic in nature. The main justifications for using it seem to be that arises from a very straightforward modification of the standard training scheme for binary neural networks, and that it has some similarity with the model of [6]. However, there are principled ways of introducing metaplasticity available in the literature [8], and one wonders whether using such synaptic models would do better.

Unfortunately, there is no comparison between different models of metaplasticity (or other learning algorithms incorporating metaplastic elements). Given that, what we can conclude the results presented here is that some form of metaplasticity is broadly useful also in the context studied here, in agreement with the general conclusions of the metaplasticity literature, but it's hard to draw strong quantitative conclusions about the merits of the particular algorithm the authors used.

The only comparison to other modern consolidation algorithms the paper offers is with elastic weight consolidation (EWC), which initially received a lot of attention in the community when it appeared, but is no longer regarded as the best algorithm for continual learning (see e.g. van de Ven & Tolias <https://arxiv.org/abs/1904.07734>). The performance on the tasks investigated seems very similar to EWC, and the main advantage of the proposed method appears to be that it doesn't need to be provided with explicit task boundaries.

In several places in the manuscript the authors point out that their algorithm is local and doesn't require a modification of the loss function. While that's certainly true, it might be worth noting that the modification of the loss function in EWC and related algorithms is really just a sum of quadratic terms, precisely one for each parameter (without mixed terms). In other words, each term certainly could be implemented locally at the level of an individual synapse also in EWC. The calculation of the coefficients of each quadratic term (e.g. the diagonal elements of the Fisher information matrix) does of course require propagating information through the network, but this procedure is no more non-local than back-propagation itself, which the authors are also using.

As an aside, it was not entirely clear to me how EWC, which in its original form is an algorithm for regularizing continuous parameters such as weights, is implemented for binary networks in the comparison the authors perform. Is it the continuous "hidden weights" that are regularized by the EWC penalty terms? This would be an additional complication in the comparison of the present algorithm with EWC, since the latter was of course not developed or optimized for binary networks, so it's unclear if its performance in this setting should be viewed as a strong benchmark.

The property of divergent internal variables shown in the analytical section of the paper reveals another problem: such divergences are precisely what good models of metaplastic synapses typically try to avoid. What it implies is that as more and more tasks are learned the system won't reach a steady state, but weights on average become more and more rigid until nothing can be learned anymore. Of course in any simulation of the system these variables won't actually have an infinite, but merely a large range, but nevertheless the steady state that is reached will have a very low degree of plasticity that doesn't support much further learning.

I think the authors should clarify whether the problem they want to solve is learning a few tasks starting from a special initial state (of small weights), or whether they want to have truly continual

learning in the sense of a steady state system that can learn a potentially infinite sequence of task in such a way that older ones are gradually forgotten, but at any point in time the more recent ones are remembered well. Much of the "continual learning" machine learning literature focuses on the former setting (in which "continual" might be a bit of a misnomer), while the metaplasticity literature typically considers the latter.

This makes a big difference, because many synaptic models that perform poorly in the steady state setting, can nevertheless often do rather well in the transient setting when they start out from a "tabula rasa" initial state. If the authors want to address the (harder) steady state problem, they would need to show results for long sequences of ever new tasks and evaluate the performance after the steady state of the hidden weight distribution has been reached. If on the other hand they only want to build a system that can learn a small number of tasks during the initial transient (of the evolution of the hidden weight distribution), that should be pointed out explicitly. In that case, perhaps even simpler methods such as judiciously scaling down the learning rate as a function of time (or task number) might do equally well, and it would make sense to compare all numerical results to the optimal learning rate annealing scheme (as mentioned for Fig. 4a).

Finally, I fear the authors might be overstating the novelty of their approach a bit by claiming e.g. that metaplasticity "has never been leveraged to mitigate catastrophic forgetting in deep neural networks". Not all authors studying this problem stress that their regularizers of the synaptic weights might be interpretable in terms of metaplasticity as the authors of the present manuscript do, but that connection has certainly appeared in the literature before (in some cases explicitly, as e.g. in Kaplanis et al. <https://arxiv.org/pdf/1802.07239.pdf>).

Despite these issues, I think that this is a nice contribution to an important line of research.

Reviewer #3 (Remarks to the Author):

The authors describe a synaptic metaplasticity model for preventing catastrophic forgetting in binarized deep neural networks. The experiments are conducted on standard, vision, supervised benchmarks.

The technique proposed is inspired by Fusi et al.'s work on (internal) synaptic consolidation states, without the requirement of task labels. Indeed, most approaches against catastrophic forgetting rely on an oracle that emits task labels.

Unfortunately, the article misses an important contribution in Zenke and Ganguli, 2017 (cited) which is that their approach already makes this assumption. In this sense, the statement in the abstract "However, such metaplastic behaviour has never been leveraged to mitigate catastrophic forgetting in deep neural networks." is incorrect.

Likewise, their method does not imply a change in the loss function. Thus, the statement "In all these techniques, the desired memory effect is enforced by changing the loss function and does not emerge from the synaptic behavior itself." is incorrect.

A few other work have taken direct inspiration from Benna and Fusi's consolidation mechanism, e.g. Kaplanis et al. "Continual Reinforcement Learning with Complex Synapses", and might be worth discussing.

Biological plausibility or spatiotemporally local computability of the gradient updates is a desired feature here. As a side note, batch normalization is not a temporally local operation as it requires computing batch statistics.

The remaining contribution is thus the application of synaptic metaplasticity to binarized neural networks. The interpretation of the hidden weights from the viewpoint of the binary weights is intriguing and novel, however.

Overall, the article is clearly written, but suffers from lack of detailed reading of previous work.

Consequently, the focus of the narrative and results is not sufficiently novel. The remaining contributions to the topic of synaptic metaplasticity is rather thin.

Response to Reviews

We would like to thank the anonymous reviewers for their time and comments, which have allowed us to improve the quality of our manuscript. We have addressed the points raised by the reviewers and revised the manuscript accordingly. The most significant additions of the revision include:

- Our approach is introduced in a more principled way, and its goals are described more explicitly.
- We introduce new benchmarks with regards to published approaches.
- We introduce a variation of our scheme using more complex synapses, inspired by recent works on metaplasticity, which can perform steady-state continual learning. This allows us to discuss the merits and challenges of using more complex metaplasticity and to precise the message of the paper.
- We have extended the discussion about locality, and its impact on hardware implementations of our scheme, in particular using nanotechnology.

In our revised manuscript, new content and sentences edited for content are marked in blue.

Reviewer #1

This paper proposes a new method to avoid catastrophic forgetting in binary neural networks (BNN). The main contribution is to first interpret the continuous weights in BNNs as a "metaplastic variable" that indicates the weights importance for remembering the past knowledge. This interpretation is then used to propose a training method which involves modification of Adam update using a function f_{meta} (line 6 and 7 in algorithm 1). An advantage of the method is that it does not require explicit task boundaries or modification of loss function. Experiments are shown on MNIST-based data (with only one experiment on CIFAR). Comparison with EWC and Random consolidation method are shown. It is not clear if the method really beats the baselines. One mathematical interpretation on a quadratic loss is also discussed.

My overall impression

(1) The method seems useful but the choices made in the algorithm are not justified well and lack a principled approach to attack the problem of continual learning.

(2) Experiments, even though incomplete, do highlight that the method could have an advantage, but the experiments are presented in a misleading way and this needs to be fixed.

We thank the reviewer for his/her review. In this revised version, we discuss the principles and the vision behind our approach much more thoroughly, we provide new experiments that provide better insight on the approach, and we have fixed the aspects identified as misleading by the reviewer.

Details

Q1 Regarding point 1: In Line 6 UW is used to make decision to perform metaplastic update. In Line 7 the function f_{meta} is used to do the update. These choices are not justified. What if we use RMSprop/SGD? Why specifically UW? What if we use some other thresholding mechanism? What about other functions for f_{meta} ? What kind of properties we need for this function? And why?

When I think about all this, I feel that the jump from neuroscience motivation to an algorithm is too quick. The paper does not have principled ways of connecting the two. This is a big issue.

Based on this comment, and a similar one from Reviewer 2, we now explain in much more detail the principles behind our approach, and its inspiration from the initial metaplasticity model. We have introduced a new section in the manuscript for this purpose: *“Interpreting the Hidden Weights of Binarized Neural Networks as Metaplasticity States”*.

We provide here a quick summary of the points raised by the reviewers (all these aspects are included in the new section):

- All experiments in this work use UW/adaptive moment estimation (Adam). Momentum-based training and root mean square propagation (RMSprop) showed equivalent results. However, pure stochastic gradient descent (SGD) leads to lower accuracy, as usually observed in binarized neural networks, where momentum is an important element to stabilize training.
- The specific choice of this set of functions f is motivated by the conceptual properties that we want our model to share with the cascade metaplasticity model. The switching ability of a binary weight should decrease exponentially the farther the hidden weight is from zero. On the other hand, the switching ability should remain unaffected when the hidden weight is close to zero, making the learning process of such weights analogous to the training of a conventional binarized neural network. These considerations led naturally to the functions used in the article, but others can be used. We also now included in the article an alternative simplified and irreversible thresholding mechanism, where the function f_{meta} is constant until a threshold value and then strictly zero (Suppl. Note 4), which we discuss in the Discussion section of the article. This type of f function also allows continual learning, with a modest accuracy degradation with regards to our initial proposal.

Another, very different, aspect motivating the choice of our f_{meta} function is that this type of continuous function could be implemented quite naturally with memristive nanodevices, offering a lead for highly efficient neuromorphic hardware of continual learning. This prospect of our work toward nanotechnology (which was in fact our initial motivation) is now discussed in detail in the Discussion section:

“In particular, the fact that metaplastic approaches build on synapses with rich behaviour resonates with the progress of nanotechnologies, which can provide compact and energy-efficient electronic devices able to mimic [12/18] neuroscience-inspired models, employing “memristive” technologies [28–31]. Many works in nanotechnologies have shown that a single nanometer-scale device can provide metaplastic behaviour [32–36]. The metaplasticity features of these nanodevices vary greatly depending on their underlying physics and technology, but their complexity is analogous to our proposal here. Typically, metaplasticity occurs by transforming the shape of a conductive filament in a continuous fashion. These changes make the device harder to program, and therefore provide a feature than can be analogous to our continuous metaplasticity function f_{meta} .”

Q2 Regarding Point 2: Looking at the left column in Figure 2, the results seems very good. As we increase m , the forgetting is avoided. Great! But looking into the other results I found this results to be misleading. In Table 1, Metaplasticity is in bold, but it doesn't beat EWC. In fact, in Supplementary Figure 1, EWC is consistently better on a variety of architectures. There the performance is reported for all 10 tasks, while in Table 1 it is only reported for 6 tasks. My conclusion is that the results are presented in a misleading way and this could have been avoided easily.

To clarify the paper, we have brought Supplementary Figure 1 as a new Figure 3 within the main paper, and included a more detailed discussion of its content in the body text. Our message is not that our approach (no longer in bold in Table 1) outperforms EWC, but that a local technique, without adding any terms to the loss, and without using any task boundaries can bring results reasonably close to EWC. The absence of task boundaries allows new possibilities such as stream learning presented in the paper. The locality, the absence of modifications of the loss and of task boundaries also makes our technique particularly amenable for hardware neuromorphic implementations, in a wide range of settings. We have clarified this message throughout the whole revised manuscript.

Q3 Some other fixes would be

- including more datasets and show consistent performance boost. MNIST based tasks are usually not that hard, and it is better to do this on CIFAR-10, Omniglot etc.

- include better baselines (EWC is one but other methods can also be applied).

We have added three new benchmarks to the paper, which each brings a new perspective on our results.

- Comparison with synaptic intelligence of (Zenke and Ganguli [23]) in the new Suppl. Note 3 and Suppl. Fig. 1.

The synaptic intelligence approach puts a lot of the computation within the synapse, similarly to our approach, but retains explicit task boundaries, which our approach does not need.

“We also compare our approach with “synaptic intelligence” introduced in [23] in Supplementary Fig.1. This approach features task boundaries as elastic weight consolidation, but can perform most operations locally, bringing it closer to biology, which retaining near-equivalent accuracy to elastic weight consolidation [23]. Contrary to elastic weight consolidation, synaptic intelligence does not adapt well to a binarized neural network: the importance factor involving a path integral cannot be computed in a natural manner using binarized weights (see Suppl. Note 3), leading to poor performance (Supplementary Fig.1(b)). On the other hand, synaptic intelligence applied to full precision neural networks requires less synapses than our binarized approach for equivalent accuracy (Supplementary Fig.1(a)), as binarized neural networks always require more synapses than full precision ones to reach equivalent accuracy [16,26]. These results highlight that the major motivation for our approach is not the raw accuracy – our technique approaches but does not match the accuracy of task-separated approaches – but the possibilities allowed by the absence of task boundaries and the use of a purely local consolidation approach. Example of such possibilities are detailed in the next sections.”

- Learning rate schedulers (new supplementary Note 7)

As a control of the fact that our approach truly identifies important synapses, we have studied the effect of scaling down uniformly the learning rate as a function of the task number for a wide range of scaling and initial learning rates and present these results in the new supplementary Note 7. We concluded that some settings give memory effects, but are far from being as effective as our metaplastic approach. Uniformly scaling down the learning rate for all synapses at once does not provide a wide range of synaptic plasticity where important synapses are consolidated, and less important ones are more plastic.

In the body text, we also included the best learning rate decay result as a new column in Table 1, as well as the new explanation:

“We also perform a control experiment by decreasing the learning rate between each task. The initial learning rate is 5×10^{-3} and is divided by ten for each new task. (This schedule provided the best results. More learning rates and dividing factors are extensively investigated in Supplementary Note 7). This technique achieves some memory effects but is not as effective as other consolidation methods: uniformly scaling down the learning rate for all synapses at once does not provide a wide range of synaptic plasticity where important synapses are consolidated and less important ones are more plastic.”

- Comparison with a more advanced metaplasticity model (new Suppl. Note 8 and new Suppl. Fig. 4)

We introduced a final benchmark to compare our work, which focuses on a simple metaplasticity rule, with a more complicated metaplasticity approach such as the one Benna and Fusi in [8]. We introduce a new plasticity rule for binarized neural networks, inspired by [8] (in the same way that the initial rule was inspired by [6]). We show that this type of synapses can allow achieving steady-state continual learning, and this allows us to provide an extended discussion on the merits and drawbacks of sophisticated metaplastic synapses versus simpler ones.

As this new work allows responding very directly to comment Q6 of the reviewer #2, we describe it in details when answering to this comment.

Q4 Two other points that I found problematic

- Why does this not work for full precision? The explanation in line 135, says that the weights contain "a trace of the history of the network updates that is relevant for memory effect". The weights in the full precision also contain the history, so then the principle should also (somehow) transfer to "full precision".

We have overhauled the explanation about this essential question. We have also added the new Supplementary Note and the associated new Supplementary Figure 3 (also reproduced here), which allow understanding this aspect visually.

The sentence formerly at line 135 now reads:

“Supplementary Fig.3 illustrates this difference visually. In a full precision neural network, “important” weights for a task converge to an optimum value minimizing the loss. By contrast, in a binarized neural network, when a binarized weight has stabilized to its optimum value, its hidden weight keeps increasing, thereby clearly indicating that the synapse should be consolidated. At the end of the paper, we provide a deeper mathematical interpretation of this intuition.”

Supplementary Figure 3. Comparison between a binarized model hidden weights and full precision weights. Weights trajectories in the case of a 2-D optimization task. The hidden weights of a binary model (a) are an accumulation of gradients evaluated in the binary values (black dot). (b) In the case of a full precision model, the weight values cannot directly be used for weight consolidation.

Q5 - The abstract says "we highlight a connection between metaplasticity ... and BNN", then the main text changes this to "we interpret the hidden weights ... as metaplastic variable". The former is misleading. The paper's method is motivated from the working mechanism of brain, and in the weights in BNN can then just be interpreted this way.

We have corrected this sentence of the abstract, which now reads:

"In this work, we realize that the hidden weights that are used by binarized neural networks, a low-precision version of deep neural networks, can be interpreted as a metaplasticity variable."

Reviewer #2

This paper approaches the problem of catastrophic forgetting in deep, binary neural networks (BNN) using a version of the BNN training algorithm modified to introduce a form of synaptic metaplasticity. The article is well written and shows some interesting empirical results in support of the usefulness of metaplasticity in continual learning.

We thank the reviewer for his/her review.

I have some comments/concerns:

Q1. The authors stress the importance of power-law (rather than exponential) forgetting in the introduction. Was there any evidence of power-law forgetting in their network?

As the paper focuses on practical settings with a reasonable number of tasks, it is difficult to prove a power law in a rigorous manner. To avoid any confusion, the works of Stefano Fusi on power law forgetting are now clearly explained in the new *"Interpreting the Hidden Weights of Binarized Neural Networks as Metaplasticity States"* section, so that it is clear that power law forgetting constitutes the inspiration behind our approach.

Q2. The synaptic metaplasticity model used in this paper is rather ad hoc and heuristic in nature. The main justifications for using it seem to be that arises from a very straightforward modification of the standard training scheme for binary neural networks, and that it has some similarity with the model of [6]. However, there are principled ways of introducing metaplasticity available in the literature [8], and one wonders whether using such synaptic models would do better.

This comment was very important to us, and we have substantially overhauled the paper to address it, following two directions. First, we have introduced a new section, *“Interpreting the Hidden Weights of Binarized Neural Networks as Metaplasticity States,”* to present the principled approach of our work and bridge the gap between the neuroscience motivation and our algorithm. We highlight which parts of [6] are the principles that we are reproducing, and what had to be adapted for binarized neural networks.

Second, we wanted to contrast our work, which indeed focuses on a simple metaplasticity rule, with more complicated approaches such as the one presented in [8]. We introduce a new plasticity rule for binarized neural networks, inspired by [8] (in the same way that the initial rule was inspired by [6]). We show that this type of synapses can allow achieving steady-state continual learning, and this allows us to provide an extended discussion on the merits and drawbacks of sophisticated metaplastic synapses versus simpler ones.

As this new work also allows responding very directly to comment Q6 of the reviewer, we describe it in detail when answering to comment Q6.

Q3. Unfortunately, there is no comparison between different models of metaplasticity (or other learning algorithms incorporating metaplastic elements). Given that, what we can conclude the results presented here is that some form of metaplasticity is broadly useful also in the context studied here, in agreement with the general conclusions of the metaplasticity literature, but it’s hard to draw strong quantitative conclusions about the merits of the particular algorithm the authors used.

The only comparison to other modern consolidation algorithms the paper offers is with elastic weight consolidation (EWC), which initially received a lot of attention in the community when it appeared, but is no longer regarded as the best algorithm for continual learning (see e.g. van de Ven & Tolias <https://arxiv.org/abs/1904.07734>). The performance on the tasks investigated seems very similar to EWC, and the main advantage of the proposed method appears to be that it doesn’t need to be provided with explicit task boundaries.

Our revision addresses this point with new results, following two directions. Overall, these new results allowed us to precise the claims of the papers.

- We now include a comparison with another continual learning approach (Zenke et al.) that also emphasizes computations done locally by synapses, but retains explicit task boundaries. This is presented in the new Supplementary Fig.1 and Suppl. Note 3, and discussed in the body text:

“We also compare our approach with the “synaptic intelligence” introduced in [23] in Supplementary Fig.1. This approach features task boundaries as elastic weight consolidation, but can perform most operations locally, bringing it closer to biology, while retaining near-equivalent accuracy to elastic weight consolidation [23]. Contrary to elastic weight consolidation, synaptic intelligence does not adapt well to a binarized neural network: the importance factor involving a path integral cannot be computed in a natural manner using binarized weights (see Suppl. Note 3), leading to poor performance (Supplementary Fig.1(b)). On the other hand, synaptic intelligence applied to full precision neural

networks requires less synapses than our binarized approach for equivalent accuracy (Supplementary Fig.1(a)), as binarized neural networks always require more synapses than full precision ones to reach equivalent accuracy [16,26]. These results highlight that the major motivation for our approach is not the raw accuracy – our technique approaches but does not match the accuracy of task-separated approaches – but the possibilities allowed by the absence of task boundaries and the use of a purely local consolidation approach. Example of such possibilities are detailed in the next sections.”

- To compare with more recent works on metaplasticity, as already mentioned in Q2, we introduced a new plasticity rule for binarized neural networks, inspired by the recent works of Benna and Fusi [8] (in the same way that the initial rule was inspired by [6]). We show that this type of synapses can allow achieving steady-state continual learning, and this allows us to provide an extended discussion on the merits and drawbacks of sophisticated metaplastic synapses versus simpler ones. *As this new work allows responding very directly to comment Q6 of the reviewer, we describe it in detail when answering to comment Q6.*

Q4 In several places in the manuscript the authors point out that their algorithm is local and doesn't require a modification of the loss function. While that's certainly true, it might be worth noting that the modification of the loss function in EWC and related algorithms is really just a sum of quadratic terms, precisely one for each parameter (without mixed terms). In other words, each term certainly could be implemented locally at the level of an individual synapse also in EWC. The calculation of the coefficients of each quadratic term (e.g. the diagonal elements of the Fisher information matrix) does of course require propagating information through the network, but this procedure is no more non-local than back-propagation itself, which the authors are also using.

The fact that our algorithm does not require a modification of the loss has two specific benefits that approaches such as EWC do not possess:

- Our work has no task boundaries, allowing the deployment of continual learning in new embedded contexts, such as the stream learning presented in the article.
- Our approach does not depend on backpropagation, and can be applied to many different approaches of learning.

These points are now more clearly stated in the paper. We have also largely extended the discussion of the question of locality, and provide a more precise and deeper view on this question:

“Additionally, taking inspiration from the metaplastic behaviour of actual synapses of the brain resulted in a strategy where the consolidation is local in space and time. This makes this approach particularly suited for artificial intelligence dedicated hardware and neuromorphic computing approaches, which can save considerable energy by employing circuit architectures optimized for the topology of neural network models, and therefore limiting data movements [37].

The fact that our metaplasticity approach is entirely local should be put into perspective into the non-local aspects of the overall learning algorithms. First, all our simulations use batch-normalization, as it is known to efficiently stabilize the training of binarized neural networks [12, 13]. Batch-normalization is not, however, a fundamental element of the scheme. Normalization technique that do not involve batches, such as instance normalization [38], layer normalization [39], or online normalization [40] provide more hardware-friendly alternatives. More profoundly, error backpropagation itself is of course non-local. Currently, multiple efforts aim at developing more local alternatives to backpropagation [41–43], or at relying on directly bioinspired learning rules [44, 45]. We have seen that alternative approaches of the literature to overcome catastrophic forgetting typically rely on the use of additional terms in the loss, are therefore strongly tied to the use of error backpropagation.”

Q5 As an aside, it was not entirely clear to me how EWC, which in its original form is an algorithm for regularizing continuous parameters such as weights, is implemented for binary networks in the comparison the authors perform. Is it the continuous “hidden weights” that are regularized by the EWC penalty terms? This would be an additional complication in the comparison of the present algorithm with EWC, since the latter was of course not developed or optimized for binary networks, so it’s unclear if its performance in this setting should be viewed as a strong benchmark.

In fact, EWC adapts extremely well to binarized neural networks! All the elements of EWC can naturally be computed using the binarized weights, and not the hidden ones. In the surrogate loss of EWC,

$$\sum_i \frac{\lambda}{2} F_i (\theta_i - \theta_{A,i}^*)^2$$

we can use directly the binarized weights as parameters θ and θ^* . The diagonal elements of the Fischer information matrix (F_i) can also solely be estimated using the binarized weights. This way, we really consolidate the binary weights and not the hidden real ones (which would indeed lead to bad results).

We have now clarified the Methods section,

“For the control, elastic weight consolidation is applied to binarized neural networks by consolidating the binary weights (and not the hidden weights as the response of the network is determined by the binary weights): both the surrogate loss term, and the Fisher information estimates are computed using the binary weight values”

and added an explicit reference to the Methods section in the main body for adaptation details.

Q6 The property of divergent internal variables shown in the analytical section of the paper reveals another problem: such divergences are precisely what good models of metaplastic synapses typically try to avoid. What it implies is that as more and more tasks are learned the system won’t reach a steady state, but weights on average become more and more rigid until nothing can be learned anymore. Of course in any simulation of the system these variables won’t actually have an infinite, but merely a large range, but nevertheless the steady state that is reached will have a very low degree of plasticity that doesn’t support much further learning.

I think the authors should clarify whether the problem they want to solve is learning a few tasks starting from a special initial state (of small weights), or whether they want to have truly continual learning in the sense of a steady state system that can learn a potentially infinite sequence of task in such a way that older ones are gradually forgotten, but at any point in time the more recent ones are remembered well. Much of the “continual learning” machine learning literature focuses on the former setting (in which “continual” might be a bit of a misnomer), while the metaplasticity literature typically considers the latter.

This makes a big difference, because many synaptic models that perform poorly in the steady state setting, can nevertheless often do rather well in the transient setting when they start out from a “tabula rasa” initial state. If the authors want to address the (harder) steady state problem, they would need to show results for long sequences of ever new tasks and evaluate the performance after the steady state of the hidden weight distribution has been reached. If on the other hand they only want to build a system than can learn a small number of tasks during the initial transient (of the evolution of the hidden weight distribution), that should be pointed out explicitly.

We have significantly overhauled the paper to address this very important point, and we thank the reviewer for raising it. This allowed us to introduce new results, and to provide an extensive discussion of the merits of simple versus complicated metaplasticity.

In summary, our simple metaplastic approach is indeed appropriate for the traditional non-steady-state continual learning situation: when the network reaches its capacity, new tasks are not learned with high accuracy. To confront this with the use of more complex metaplasticity approaches, we introduced in Suppl. Note 8, a new plasticity rule for binarized neural networks, inspired by the recent works of Benna and Fusi [8] (in the same way that the initial rule was inspired by [6]). We show that this type of synapses can allow achieving steady-state continual learning, and this allows us to provide an extended discussion on the merits and drawbacks of sophisticated metaplastic synapses versus simpler ones. Highly complex synapses provide more continual learning features, but have an implementation cost. We highlight that the type of simple metaplastic synapses introduced in our paper is the ideal level for implementation with nanodevices — several technologies have been shown to exhibit intrinsic effects similar to metaplasticity. This was actually the main initial motivation of our work, although it was not emphasized in the initial version of the manuscript.

Here is the new discussion introduced in the body text concerning these new results, as well the associated new Suppl. Fig. 4 (we do not reproduce the whole associated Suppl. Note 8 here as it is quite long):

“A distinctive aspect of continual learning approaches is their behavior when the neural network reaches its capacity in terms of number of tasks. The behavior in the case of our approach can be anticipated from the mathematical interpretation in the previous section: when all hidden weights have started to diverge, i.e., are consolidated for a given task, no weights should be able to learn new tasks. The consequence of this situation is well seen in Suppl. Fig.4(b): when learning ten permuted MNIST tasks, the last task has reduced accuracy, while the first trained tasks retain their original accuracy. This behavior fits well with a large section of the literature on continual learning, multitask learning, where the goal is to learn a given number of tasks [27]. Suppl. Fig.2 also highlights the relative definitive nature of synaptic consolidation in our approach. We implemented a variation, where the metaplasticity function reaches a hard zero after a given threshold. We see that the performance on the ten permuted MNIST tasks is only modestly reduced by this change..

This behavior also differentiates our approach from the brain, where a more natural behavior for most networks would be to forget the earliest trained tasks, and replace them with the newly trained ones. In recent years, the literature about metaplasticity has aimed at reproducing this behavior, i.e., a type of “steady-state” continual learning [8, 11]. This recent literature can therefore provide leads to provide our network with this behavior. In particular Benna et al proposed a metaplasticity model where synapses feature a network of different elements, which all evolve at different time scales [8]. This model can feature a sophisticated memory effect, and one work successfully used this type of the synapses in the context of an elementary continual reinforcement learning task related to the Cart-Pole problem [11].

We found that directly applying the metaplasticity rule of [8] in our context does not yield proper memory effects. The explanation stems from the specificity of deep networks: in [8], synaptic updates occur following randomly presented patterns, in an independent and identically distributed fashion. In our continual learning situation, sequential synaptic updates follow the gradient of a loss function and are therefore highly correlated. However, the rule of [8] can still be used as an inspiration to allow steady-state continual learning in our approach. In Supplementary Note 8 and the associated Supplementary Fig.4, we provide a learning rule where synapses also feature a network of elements evolving at different time scale adapted for the training of binarized neural networks, leading to a natural forgetting of tasks trained a long time ago when new tasks are trained. Our adaptation consists in modulating the flow between hidden variables, an idea suggested as a perspective in [11] as a way to bridge the gap between conventional continual learning methods and neuroscience based

approaches. We can see in Suppl. Fig.4(c) that in this case, when training ten permuted MNIST tasks, the last trained task features the highest accuracy, while the accuracy of the first trained tasks starts to decrease.

This discussion highlights an interplay between the level of continual learning feature and of synaptic complexity. Highly complicated synapses, featuring many equations and hyperparameters, as the ones of [8, 11] or the one that we just introduced, can achieve advanced continual learning behaviors. Biology appears to have followed this path, as synapses seem to exhibit highly complex metaplastic behaviors [10]. For artificial system, the richness of highly complex synapses need to be counter-balanced with their implementation cost. This discussion is natural for software implementations of metaplasticity, and also exists for hardware. In particular, the fact that metaplastic approaches build on synapses with rich behaviour resonates with the progress of nanotechnologies, which can provide compact and energy-efficient electronic devices able to mimic [12/18] neuroscience-inspired models, employing “memristive” technologies [28–31]. Many works in nanotechnologies have shown that a single nanometer-scale device can provide metaplastic behaviour [32–36]. The metaplasticity features of these nanodevices vary greatly depending on their underlying physics and technology, but their complexity is analogous to our proposal here. Typically, metaplasticity occurs by, in a continuous fashion transforming the shape of a conductive filament. These changes make the device harder to program, and therefore provide a feature than can be analogous to our continuous metaplasticity function f_{meta} . On the other hand, the complicated version of Suppl. Note 8 would be highly challenging to implement with a single nanodevice, based on the current state of nanotechnologies, as these metaplasticity models require many different states with different time dynamics. Our proposal could therefore be an outstanding candidate for nanotechnological implementations, as it provides rich features at the network level, while remaining compatible with the constrains of technology.”

Supplementary Figure 4. Extension to Stationary Weight Distribution (a)Schematic of a more complex synapse model. (b)Test accuracies of ten tasks for a metaplastic BNN as introduced in our main article

with $m=1.35$ and two hidden layers of 4,096 units. The tasks are learned until no further learning can be done (task #8 to #10 are not properly learned). (c) Same architecture but with our new algorithm with four hidden variables. The model is still able to learn several tasks sequentially but older tasks are gradually forgotten and new tasks can always be learned. (d) Trajectories of the hidden variables in function of training iterations. The deeper the hidden variable, the slower and smoother it behaves, providing a cleaner signal for consolidation. (e) Distribution of the hidden variables after learning 10 tasks. Unlike the distribution presented in the body text, hidden weights do not accumulate to ever increasing values and new tasks can always be learned.

Q7 In that case, perhaps even simpler methods such as judiciously scaling down the learning rate as a function of time (or task number) might do equally well, and it would make sense to compare all numerical results to the optimal learning rate annealing scheme (as mentioned for Fig. 4a).

To answer this question, we have studied the effect of scaling down the learning rate uniformly as a function of the task number for a wide range of scaling and initial learning rates, and present these results in a new supplementary Note 7. We concluded that some settings give memory effects, but are far from being as effective as our metaplastic approach. Uniformly scaling down the learning rate for all synapses at once does not provide a wide range of synaptic plasticity where important synapses are consolidated, and less important ones are more plastic.

In the body text, we also included the best learning rate decay as a new column in Table 1, as well as the new explanation:

“We also perform a control experiment by decreasing the learning rate between each task. The initial learning rate is 5×10^{-3} and is divided by ten for each new task. (This schedule provided the best results. More learning rates and dividing factors are extensively investigated in Supplementary Note 7). This technique achieves some memory effects but is not as effective as other consolidation methods: uniformly scaling down the learning rate for all synapses at once does not provide a wide range of synaptic plasticity where important synapses are consolidated and less important ones are more plastic.”

Q8 Finally, I fear the authors might be overstating the novelty of their approach a bit by claiming e.g. that metaplasticity “has never been leveraged to mitigate catastrophic forgetting in deep neural networks”. Not all authors studying this problem stress that their regularizers of the synaptic weights might be interpretable in terms of metaplasticity as the authors of the present manuscript do, but that connection has certainly appeared in the literature before (in some cases explicitly, as e.g. in Kaplanis et al. <https://arxiv.org/pdf/1802.07239.pdf>).

We changed the sentence of the abstract to a more indisputable: “However, such “metaplastic” behaviours do not transfer directly to mitigate catastrophic forgetting in deep neural networks. “, and discuss more extensively the work of Kaplanis and Zenke in our state of the art.

Despite these issues, I think that this is a nice contribution to an important line of research.

We thank the reviewer for his/her appreciation of our work.

Reviewer #3

The authors describe a synaptic metaplasticity model for preventing catastrophic forgetting in binarized deep neural networks. The experiments are conducted on standard, vision, supervised benchmarks.

The technique proposed is inspired by Fusi et al.'s work on (internal) synaptic consolidation states, without the requirement of task labels. Indeed, most approaches against catastrophic forgetting rely on an oracle that emits task labels.

We thank the reviewer for his/her review.

Q1 Unfortunately, the article misses an important contribution in Zenke and Ganguli, 2017 (cited) which is that their approach already makes this assumption. In this sense, the statement in the abstract "However, such metaplastic behaviour has never been leveraged to mitigate catastrophic forgetting in deep neural networks." is incorrect.

Likewise, their method does not imply a change in the loss function. Thus, the statement "In all these techniques, the desired memory effect is enforced by changing the loss function and does not emerge from the synaptic behavior itself." is incorrect.

We changed the abstract accordingly to *"However, such "metaplastic" behaviours do not transfer directly to mitigate catastrophic forgetting in deep neural networks."*

We also discussed the Zenke paper more thoroughly in the introduction. We define in a much more precise manner how the penalty is introduced in the loss function, and in which aspect this approach differs from ours:

"Finally, in [23], the consolidation strategy consists in computing an importance factor based on path integral. This last approach is the closest to the biological models of metaplasticity, as all computations can be performed at the level of the synapse, and the importance factor is therefore reminiscent of a metaplasticity parameter. However, in all these techniques, the desired memory effect is enforced by optimizing a loss function with a penalty term which depends on the previous optimum, and does not emerge from the synaptic behaviour itself. This aspect requires a very formal separation of the tasks – the weight values at the end of task training need to be stored – and makes these models still largely incompatible with the constraints of biology and embedded contexts. The highly non-local nature of the consolidation mechanism also makes it difficult to implement in neuromorphic-type hardware."

We specified the meaning of our sentence by changing it to: *"In all these techniques, the desired memory effect is enforced by optimizing a loss function with an penalty term which depends on the previous optimum, and does not emerge from the synaptic behaviour itself."*

Q2 A few other work have taken direct inspiration from Benna and Fusi's consolidation mechanism, e.g. Kaplanis et al. "Continual Reinforcement Learning with Complex Synapses", and might be worth discussing.

The introduction and the discussion of the paper now clearly highlight the work of Kaplanis (Ref. 11).

In addition, to discuss extensively how our work compares with the one of Benna and Fusi (Ref 8), we have performed new work. We introduce a new plasticity rule for binarized neural networks, inspired by [8] (in the same way that the initial rule was inspired by [6]). We show that this type of synapses can allow achieving steady-state continual learning, and this allows us to provide an extended discussion on the merits and drawbacks of sophisticated metaplastic synapses versus simpler ones, an important addition to the paper, which clarifies its goal.

As this new work allowed responding very directly to comment Q6 of the reviewer #2, we described it in details when answering to comment Q6.

Q4 Biological plausibility or spatiotemporally local computability of the gradient updates is a desired feature here. As a side note, batch normalization is not a temporally local operation as it requires computing batch statistics.

We have largely extended the discussion of the question of locality, and provide a more precise and deeper view on this question:

“Additionally, taking inspiration from the metaplastic behaviour of actual synapses of the brain resulted in a strategy where the consolidation is local in space and time. This makes this approach particularly suited for artificial intelligence dedicated hardware and neuromorphic computing approaches, which can save considerable energy by employing circuit architectures optimized for the topology of neural network models, and therefore limiting data movements [37].

The fact that our metaplasticity approach is entirely local should be put into perspective into the non-local aspects of the overall learning algorithms. First, all our simulations use batch-normalization, as it is known to efficiently stabilize the training of binarized neural networks [12, 13]. Batch-normalization is not, however, a fundamental element of the scheme. Normalization technique that do not involve batches, such as instance normalization [38], layer normalization [39], or online normalization [40] provide more hardware-friendly alternatives. More profoundly, error backpropagation itself is of course non-local. Currently, multiple efforts aim at developing more local alternatives to backpropagation [41–43], or at relying on directly bioinspired learning rules [44, 45]. We have seen that alternative approaches of the literature to overcome catastrophic forgetting typically rely on the use of additional terms in the loss, are therefore strongly tied to the use of error backpropagation. On the other hand, as our metaplasticity approach is entirely synaptic-centric, it is largely agnostic to the learning rule, and should be adaptable to all these emerging learning approaches.”

Q5 The remaining contribution is thus the application of synaptic metaplasticity to binarized neural networks. The interpretation of the hidden weights from the viewpoint of the binary weights is intriguing and novel, however.

Overall, the article is clearly written, but suffers from lack of detailed reading of previous work. Consequently, the focus of the narrative and results is not sufficiently novel. The remaining contributions to the topic of synaptic metaplasticity is rather thin.

We thank the reviewer for acknowledging the novelty of our study. We hope that our new results and discussion very significantly improve the depth and the impact of our initial submission.

Reviewer #1 (Remarks to the Author):

Thanks for revising the manuscript. Reading through your responses, I found them reasonable and adequate, and I thank you for that.

- The interpretation of "latent weights" as "metaplastic variable" is reasonable, even though it is just an intuitive guess.
 - Compared to EWC, the advantages of no task boundary and local (biologically motivated) updates is also reasonable.
 - Addition of three benchmarks is good, although it is only done for one dataset MNIST which does not improve the experiments reasonably. I had
- So overall this is an improvement to the submitted version, although there are still two issues that need to be fixed.

First issue: The abstract again is misleading.

- It says the you present "a training technique that *prevents* catastrophic forgetting". This is not true - at best it *reduces* the forgetting.
- The abstract does not clearly say that you method does not really beat existing methods (EWC) rather enables you to perform similarly (but sometime worse as shown in Fig. 3) while not requiring task boundaries or modifying the loss function. I don't understand why the authors would want to not clearly write about their finding, and rather make statements that are not true.
- Also the bit about "we realize that the hidden weights... can be interpreted as a metaplastic variable" is also misleading. The statement conveys that the "the hidden weights *are* indeed metaplastic variables and the authors seem to realize it. It might seem like word play, but you can simply say "we interpret the hidden weights as metaplastic variable and modify it to reduce overfitting during training".

Second issue: Add two more datasets (CIFAR-10, CIFAR-100) to Fig. 2

- add CIFAR-10, CIFAR-100 to the experiment in Fig. 2. This was asked in the reviews to include more datasets, but this is ignored in the revised version.

Here is a helpful bit. I found a reference that might be useful to establish the connection between hidden weights.

>Training Binary Neural Networks using the Bayesian Learning Rule
Xiangming Meng, Roman Bachmann, Mohammad Emtiyaz Khan

They interpret latent weights as the natural parameter of Bernoulli distribution. They use the distribution to avoid catastrophic forgetting. The gradients in their case are scaled by a quantity which is related to tanh, but is different from yours.

This paper is recently published, so I am not implying that you have to compare, but I think there are similarities that might help you improve your explanations.

Reviewer #2 (Remarks to the Author):

The authors have substantially improved the paper and addressed the majority of my concerns. Some further clarification on the following points might be helpful:

The authors now state clearly that the goal of the primary model in this manuscript is to learn a small number of tasks during the transient period in which the weight distribution changes from a tightly concentrated (around zero) initial state to a wide distribution in which many weights will effectively be frozen, and further plasticity is unlikely (i.e. not a plastic steady state situation). The system thus has a finite capacity for learning tasks sequentially. How high is this capacity relative to the capacity for learning multiple task simultaneously? In particular, how many more tasks could be learned by the same network with interleaved training with/without metaplastic synapses (at say 80 or 90% performance) than in the sequential (continual learning) setting? Such a baseline capacity would be a valuable addition to e.g. the plots in Figure 3 (which by the way could use a legend to associate the colors with different network sizes). This would essentially quantify

the reduction in capacity the system suffers due to the temporal correlations of the input training examples.

I appreciate that the authors have put some work into investigating a more complex model with multiple timescales in Suppl. Note 8, which is based on [8] with the binarization of the shortest timescale variable as described in suppl. material of that paper. However, the way the present manuscript deals with this complex model raises a number of questions:

- The authors say that the model of [8] "does not yield proper memory effects" in the setting they study. A little more detail would be helpful here. In what way did this model underperform and why? The correlations between successive plasticity steps might well play a role here as the authors point out, but by itself this doesn't constitute an explanation. Could this simply be a consequence of the choice of the number of internal variables and their timescales? Suppl. Fig. 4d suggests that perhaps the slowest timescale of the model was chosen to be too short for the slowest variable to aggregate information across the training on all the tasks to be learned.
- The authors then modify this model such as to make it more similar to their single variable model by introducing f_{meta} and feedback from the slowest to the fastest variable with coefficient α . The motivations for these modifications are briefly mentioned in Suppl. Note 8, but without knowing how the original complex model failed it's not entirely clear why these are deemed necessary. E.g. how does this address the issue of correlations in the plasticity steps? This would be an opportunity for the authors to clearly state what computational benefits they derive from each of the modifications they introduce.
- Since the modified complex model retains the leak term for the slowest variable, it may technically be true that this can be called a steady state model. However, due to the introduction of f_{meta} , the slowest variable might still get stuck in a state of large absolute value for long periods of time, just like the hidden variable could get stuck in the main model of this manuscript. We can't tell from Suppl. Fig 4 whether this is a problem or not, because it doesn't show results in the steady state regime. If the authors want to call this a steady state metaplasticity model, it should really be tested in the steady state regime, i.e. by learning a long sequence of tasks and only evaluating the performance once the distributions of weights and internal variables no longer change.

Some points in the discussion could be enhanced in clarity and scope: E.g. while the present model may have the right level of complexity for nano-device applications (but note that multi-timescale synapses with fewer internal states might also be suitable for this purpose DOI: 10.1109/ACSSC.2017.8335630), implementation costs matter in the biological brain as well. Presumably evolution would not have chosen to create complex synapses if there was no computational benefit.

I'm still not entirely comfortable with the way the novelty of the present contribution is described. I understand that the authors arrived at their model by considering BNNs, and it's perfectly fine to tell that story, but it should be acknowledged that the resulting synaptic model is very similar to what is typically called the "multi-state model with binary readout" or "serial model" in the literature (see e.g. <https://papers.nips.cc/paper/4872-a-memory-frontier-for-complex-synapses.pdf> and references therein). This simple way of combining a quasi-continuous internal variable with binary synaptic weights has been studied at least since 1986 (DOI: 10.1103/physreva.34.2571). The present model differs in their choice of the metaplasticity function f_{meta} , but it is not obvious that there is anything profound about that particular choice. My impression is that the main novelty of the present manuscript lies not so much in the synaptic model itself, but in investigating its performance in the context of continual learning in multi-layer networks.

The authors are also still heavily stressing the advantage of the local processing at the synapse via metaplasticity and of not having to modify a cost function. While I completely agree that this is beneficial in principle, I think for the present study this is merely a hypothetical advantage, since the training here anyway relies on minimizing a cost function via back-propagation. For this advantage to actually be important in practice, the training would have to occur using local learning rules (or using supervisory feedback with only a limited bandwidth). My recommendation would be to de-emphasize this point in favor of emphasizing that the model works in the absence of explicit task boundaries. Even though these two issues are related in a number of previous

models they are logically separate, since one can imagine modifying the loss even without explicit task boundaries.

Reviewer #3 (Remarks to the Author):

The manuscript offers an appealing method for catastrophic forgetting that may be promising for implementation in neuromorphic hardware and resistive switching devices. The combination of experiments and mathematical interpretation is interesting and promising.

A key experiment is still missing, however: A mechanisms against catastrophic forgetting is useful mainly if it performs better than a network partitioned in a number of parts equal to the number of tasks, and each partition is trained independently from the other partitions. "Performed better" here is open for interpretation and discussion: it can consist in forward transfer, backward transfer, capacity, etc. The experiments chosen by the authors do not address this point. For demonstrating transfer, the permuted MNIST task is not a good choice. Instead, the authors may want to learn different classes in Omniglot, or learn several different digit datasets (USPS, MNIST, etc) in sequence. See also my comment on stream learning below.

The idea of a hidden variable is reminiscent also of "Network plasticity as Bayesian inference" Kappel et al, and provides another biologically inspired model of two-level plasticity. The authors may want to compare the current (deterministic) approach with the stochastic approach of Kappel et al.

p2157 "penalty term which depends on the previous optimum". One comment here (also correcting a comment in my previous review): While Zenke et al. did use a loss function based on the previous parameter set, their approach does not fundamentally necessitate a separation of tasks. However, they didn't demonstrate it: too bad for them, and good the authors of this manuscript.

Labels for the different curves (network sizes) in Figure 3 are missing.

p81169 "our weight consolidation strategy is tied specifically to the use of a binarized neural network". The implementation of the full precision network is a bit misleading: Performing updates with f_{meta} would effectively induce incorrect gradient in the full precision network. Can a different f_{meta} recover the full precision performance?

p91199: The experiments for steam learning are not informative about a key aspect of real-world learning, namely the non-iid sampling of the data. The non-random ordering in stream learning with respect to the categories is a key problem in real-world learning. In the stream learning section, the authors should instead address this, or give a good reason why "All classes are represented in each subset". What happens for instance if all shoes are learned, then all shirts, etc? One other point is that the performance of the proposed solution is highly dependent on how many updates are made, and thus the learning rate. Couldn't the gap between the baseline and the stream learning in fig 5 be reduced by increasing the learning rate?

p121301 "For artificial system" -> "For an artificial system"

Response to Reviews

We would like to thank the anonymous reviewers for their time and comments which have allowed us to improve the quality of our manuscript. We have addressed the points raised by the reviewers and revised the manuscript accordingly. The revision includes several new experiments such as class incremental learning of CIFAR-10 and CIFAR-100 features, comparison with networks trained on all tasks at once, and comparison with smaller networks trained on one task each.

In our revised manuscript, new content and sentences edited for content are marked in blue.

Reviewer #1

Thanks for revising the manuscript. Reading through your responses, I found them reasonable and adequate, and I thank you for that.

- The interpretation of "latent weights" as "metaplastic variable" is reasonable, even though it is just an intuitive guess.

- Compared to EWC, the advantages of no task boundary and local (biologically motivated) updates is also reasonable.

- Addition of three benchmarks is good, although it is only done for one dataset MNIST which does not improve the experiments reasonably. I had

So overall this is an improvement to the submitted version, although there are still two issues that need to be fixed.

We thank the reviewer for his/her comment on the previously revised version.

First issue: The abstract again is misleading.

- It says the you present "a training technique that *prevents* catastrophic forgetting". This is not true - at best it *reduces* the forgetting.

- The abstract does not clearly say that you method does not really beat existing methods (EWC) rather enables you to perform similarly (but sometime worse as shown in Fig. 3) while not requiring task boundaries or modifying the loss function. I don't understand why the authors would want to not clearly write about their finding, and rather make statements that are not true.

- Also the bit about "we realize that the hidden weights... can be interpreted as a metaplastic variable" is also misleading. The statement conveys that the "the hidden weights *are* indeed metaplastic variables and the authors seem to realize it. It might seem like word play, but you can simply say "we interpret the hidden weights as metaplastic variable and modify it to reduce overfitting during training".

We have implemented all the suggested modifications to the abstract, and they are marked in blue in the revised version. Our abstract now explicitly says that our technique reduces catastrophic

forgetting, that its accuracy only approaches existing methods with task boundaries, and that hidden weights are not in themselves metaplastic variables and need to be modified for this purpose.

Second issue: Add two more datasets (CIFAR-10, CIFAR-100) to Fig. 2

- add CIFAR-10, CIFAR-100 to the experiment in Fig. 2. This was asked in the reviews to include more datasets, but this is ignored in the revised version.

We have now included class incremental training experiments on the CIFAR-10 and CIFAR-100 datasets. They are described in detail in the new Supplementary Note 11 and the associated Supplementary Figure 8. We have also reported the most important results in Fig. 4(e-f-h) (we initially tried to incorporate them within Fig. 2, but the story of the paper flowed better if the results were included within Fig. 4).

To maintain the balance of the paper, we have simplified the description of the sequential training of the Fashion-MNIST and MNIST datasets in the body text, and moved the details in the new Supplementary Note 9 and associated new Supplementary Fig. 6. We added the following text describing the CIFAR-10 and CIFAR-100 experiments in the body text :

“Finally, we investigated a situation of class incremental learning of the CIFAR-10 (Figs. 4 (e-f)) and CIFAR-100 (Figs. 4 (g-h)) datasets. We use a convolutional neural network with convolutional layers pretrained on ImageNet, and a metaplastic classifier (see Supplementary Note 11). The classes of these datasets are divided into two subsets and trained sequentially. While in the non-metaplastic network (Figs. 4 (e-g)), the first subset of classes is forgotten rapidly when the second is trained, in the metaplastic one (Figs. 4 (f-h)), good accuracy is achieved, which remains below the one obtained with non sequentially trained classes. Better performance can be achieved if we allow the neurons to have independent thresholds for the two subsets (see Supplementary Note 11).”

Here is a helpful bit. I found a reference that might be useful to establish the connection between hidden weights. Training Binary Neural Networks using the Bayesian Learning Rule, Xiangming Meng, Roman Bachmann, Mohammad Emtiyaz Khan. They interpret latent weights as the natural parameter of Bernoulli distribution. They use the distribution to avoid catastrophic forgetting. The gradients in their case are scaled by a quantity which is related to tanh, but is different from yours.

This paper is recently published, so I am not implying that you have to compare, but I think there are similarities that might help you improve your explanations.

We thank the reviewer for this interesting suggestion. We have studied this work (now Ref. 30) and added the following sentences to the discussion:

"Other methods have been introduced to train binarized neural networks such as ²⁹ or ³⁰ and provide valuable insights to understand the specificity of binarized networks with respect to continual learning. Helweg et al.²⁹ interpret the hidden weight as inertia, which is coherent with the fact that high inertia might correspond to important weights, while Meng et al.³⁰ link the hidden weight to the natural parameter of a probability distribution over binarized weights which can be used as a relevant prior to perform continual learning."

Reviewer #2

The authors have substantially improved the paper and addressed the majority of my concerns.

We thank the reviewer for his/her comment on the previously revised version.

Some further clarification on the following points might be helpful:

The authors now state clearly that the goal of the primary model in this manuscript is to learn a small number of tasks during the transient period in which the weight distribution changes from a tightly concentrated (around zero) initial state to a wide distribution in which many weights will effectively be frozen, and further plasticity is unlikely (i.e. not a plastic steady state situation). The system thus has a finite capacity for learning tasks sequentially. How high is this capacity relative to the capacity for learning multiple tasks simultaneously? In particular, how many more tasks could be learned by the same network with interleaved training with/without metaplastic synapses (at say 80 or 90% performance) than in the sequential (continual learning) setting? Such a baseline capacity would be a valuable addition to e.g. the plots in Figure 3 (which by the way could use a legend to associate the colors with different network sizes). This would essentially quantify the reduction in capacity the system suffers due to the temporal correlations of the input training examples.

We thank the reviewer for his/her comment. We have implemented the suggested experiment where the network learns simultaneously all the tasks. The results of a metaplastic network and non-

metaplastic network are reported in two new plots in Figs. 3(c-d), using the same plotting conventions that the incremental training of Figs. 3(a-b).

When comparing these results with Fig. 3(a), we observe that the reduction in capacity due to temporal correlations is significant compared to the case where all the data is available at the same time, which adds additional insight to the paper. While all networks with varying sizes can learn ten tasks with an accuracy above 95% in the case of interleaved training, only the biggest network remains above 95% accuracy up to task 8 in the case of sequential training.

As suggested, we have also added a new legend to Fig. 3.

I appreciate that the authors have put some work into investigating a more complex model with multiple timescales in Suppl. Note 8, which is based on [8] with the binarization of the shortest timescale variable as described in suppl. material of that paper. However, the way the present manuscript deals with this complex model raises a number of questions:

- The authors say that the model of [8] “does not yield proper memory effects” in the setting they study. A little more detail would be helpful here. In what way did this model underperform and why? The correlations between successive plasticity steps might well play a role here as the authors point out, but by itself this doesn’t constitute an explanation. Could this simply be a consequence of the choice of the number of internal variables and their timescales? Suppl. Fig. 4d suggests that perhaps the slowest timescale of the model was chosen to be too short for the slowest variable to aggregate information across the training on all the tasks to be learned.

- The authors then modify this model such as to make it more similar to their single variable model by introducing f_{meta} and feedback from the slowest to the fastest variable with coefficient α . The motivations for these modifications are briefly mentioned in Suppl. Note 8, but without knowing how the original complex model failed it’s not entirely clear why these are deemed necessary. E.g. how does this address the issue of correlations in the plasticity steps? This would be an opportunity for the authors to clearly state what computational benefits they derive from each of the modifications they introduce.

Based on these comments, we have significantly extended the description of the multiple timescale schemes, based on both new empirical results and qualitative discussion.

New experimental results. We have performed an ablation study on our feedback process linking the slowest hidden variable to the first one. We let the hidden variables evolve only through the main connections and remove the feedback process: i.e., we set $\alpha = 0$ and $f_{\text{meta}} = 1$ in Suppl. Fig. 4(a).

The results are listed in the new Supplementary Table 5 for 21 values of the parameters of the synapses, covering cases with more hidden variables and/or slower time scales, as suggested by the anonymous reviewer. In all these situations, we observe some memory for Tasks 8 and 9, with varying accuracy depending on the parameter choice. However, the accuracy of Task 7 is always back to near-random guess, suggesting that catastrophic forgetting remains strong in the absence of our model modifications. This result is consistent with our interpretation that the influence of the slowest (last) hidden variable over the fastest one through the main connections is too weak to protect the first variable from the strongly correlated gradients.

Qualitative discussion. We also explain further why the model needs to be modified in order to work in the setting of deep neural networks where two different timescales are at play: the time scale within one task training and the timescale across tasks. For instance, we find that choosing a shorter slowest timescale as suggested does not help because a slow variable on the intra task timescale is not slow when observed from the inter task time scale. The purpose of our modification is to accommodate this asymmetry. The new qualitative discussion reads:

“These two additions are necessary as the dynamics of the synapses differ substantially when training binarized neural networks from the situation of⁵. In⁵, synaptic updates occur following randomly presented patterns, in an independent and identically distributed fashion. Our continual learning situation is different, because there are two distinct timescales at play: a short timescale constituted by the training iterations within one task, and a long timescale constituted by the different tasks. The slowest variable evolves slowly at the intra-task timescale but rapidly with respect to the long timescale. We introduce f_{meta} to accommodate for this timescale asymmetry. Another difference comes from the sequential synaptic updates, which follow the gradient of a loss function and are therefore highly correlated on shorter time scales. For this reason, the influence of the slowest variable on Wh_1 through the diffusion chain cannot effectively protect from the correlated gradients of the new task. We thus add a unidirectional feedback connection parameterized by (Suppl. Fig. 4 (a)) between the slowest variable and Wh_1 to provide better consolidation. The two modifications of the model allow Wh_4 to be more stable on the longer timescales of our setup, while allowing to Wh_1 react on its shorter ones.”

- Since the modified complex model retains the leak term for the slowest variable, it may technically be true that this can be called a steady state model. However, due to the introduction of f_{meta} , the slowest variable might still get stuck in a state of large absolute value for long periods of time, just like the hidden variable could get stuck in the main model of this manuscript. We can't tell from Suppl. Fig 4 whether this is a problem or not, because it doesn't show results in the steady state regime. If the authors want to call this a steady state metaplasticity model, it should really be tested in the steady state regime, i.e. by learning a long sequence of tasks and only evaluating the performance once the distributions of weights and internal variables no longer change.

We performed additional experiments to check this interesting point.

We extended the experiments of Supplementary Note 8 to 20 tasks and plot the distribution of the hidden variables for the three most recent tasks in the new Supplementary Figure 5(b). We observe that the distribution of hidden variables is truly stationary over the three most recent tasks, with a near-perfect superimposition of the distributions of the hidden weight.

Additionally, the new Suppl. Fig. 5(a) shows the accuracy of the tasks in this “truly steady-state” regime. This Figure shows another surprising effect. In the truly steady-state regime, the remembered task capacity is lower than in the regime seen in Suppl. Fig. 4. A similar effect has been reported by Benna et al in the reference suggested by the reviewer in his/her next question (DOI:

10.1109/ACSSC.2017.8335630): the steady-state capacity is lower than the capacity of the transient regime but exhibits a graceful forgetting of old tasks and continual ability to learn new tasks.

Some points in the discussion could be enhanced in clarity and scope: E.g. while the present model may have the right level of complexity for nano-device applications (but note that multi-timescale synapses with fewer internal states might also be suitable for this purpose DOI: 10.1109/ACSSC.2017.8335630), implementation costs matter in the biological brain as well. Presumably evolution would not have chosen to create complex synapses if there was no computational benefit.

Based on these comments, we have clearly highlighted that implementation costs matter in the biological brain as well, and that the fact that evolution has favored complex synapses might suggest their computational benefits. The updated text reads:

“For an artificial system, the richness of highly complex synapses needs to be counterbalanced with their implementation cost. Biology might have experienced a similar dilemma. Evolution seems to have favored synapses exhibiting highly complex metaplastic behaviors¹⁰, although simpler synapses might have been more efficient to implement, suggesting the high computational benefits of complex synapses.”

We have also cited the ACSSC paper as an alternative road for making complex synapses with nanotechnologies: *“Our proposal, as other proposals of complex synapses with multiple variables⁴¹ or stochastic behaviors⁴², could therefore be an outstanding candidate for nanotechnological implementations”*.

I’m still not entirely comfortable with the way the novelty of the present contribution is described. I understand that the authors arrived at their model by considering BNNs, and it’s perfectly fine to tell that story, but it should be acknowledged that the resulting synaptic model is very similar to what is typically called the “multi-state model with binary readout” or “serial model” in the literature (see e.g. <https://papers.nips.cc/paper/4872-a-memory-frontier-for-complex-synapses.pdf> and references therein). This simple way of combining a quasi-continuous internal variable with binary synaptic weights has been studied at least since 1986 (DOI: 10.1103/physreva.34.2571). The present model differs in their choice of the metaplasticity function f_{meta} , but it is not obvious that there is anything profound about that particular choice. My impression is that the main novelty of the present manuscript lies not so much in the synaptic model itself, but in investigating its performance in the context of continual learning in multi-layer networks.

We have edited the introduction of the paper to highlight the “multi-state model with binary readout” or “serial model” in the literature, and that the novelty of our work comes from investigating the potential and the performance of such approaches in the context of continual learning in multi-layer networks. The edited text reads: *“However, intriguingly, in the field of deep learning, binarized neural networks¹² (or the closely related XNOR-NETs¹³) have a remote connection with the concept of*

metaplasticity, also reminiscent, in neuroscience, of the multi state models with binary readout¹⁴. This connection has never been explored to perform continual learning in multi-layer networks.”

The authors are also still heavily stressing the advantage of the local processing at the synapse via metaplasticity and of not having to modify a cost function. While I completely agree that this is beneficial in principle, I think for the present study this is merely a hypothetical advantage, since the training here anyway relies on minimizing a cost function via back-propagation. For this advantage to actually be important in practice, the training would have to occur using local learning rules (or using supervisory feedback with only a limited bandwidth). My recommendation would be to de-emphasize this point in favor of emphasizing that the model works in the absence of explicit task boundaries. Even though these two issues are related in a number of previous models they are logically separate, since one can imagine modifying the loss even without explicit task boundaries.

We have performed several changes throughout the manuscript to reduce the stress on the advantage of space locality, and emphasize the absence of task boundaries instead.

“On the other hand, our approach achieves a performance similar to elastic weight consolidation for learning six permuted MNISTs with the given architecture, although unlike elastic weight consolidation the consolidation is based on an entirely local rule without changing the loss function.”

to

“On the other hand, our approach achieves a performance similar to elastic weight consolidation for learning six permuted MNISTs with the given architecture, although unlike elastic weight consolidation the consolidation does not require changing the loss function and thus does not require task boundaries.”

We also remove the part “*and the use of a purely local consolidation approach*” in the sentence

“The major motivation of our approach are the possibilities allowed by the absence of task boundaries and the use of a purely local consolidation approach.”

We also remove the part “*, and locally in space*” in the sentence

“Our consolidation strategy is carried out simultaneously with the weight update, and locally in space as consolidation only involves the hidden weights.”

Reviewer #3

The manuscript offers an appealing method for catastrophic forgetting that may be promising for implementation in neuromorphic hardware and resistive switching devices. The combination of experiments and mathematical interpretation is interesting and promising.

We thank the reviewer for his/her comment.

A key experiment is still missing, however: A mechanism against catastrophic forgetting is useful mainly if it performs better than a network partitioned in a number of parts equal to the number of tasks, and each partition is trained independently from the other partitions. "Performed better" here is open for interpretation and discussion: it can consist in forward transfer, backward transfer, capacity, etc. The experiments chosen by the authors do not address this point. For demonstrating transfer, the permuted MNIST task is not a good choice. Instead, the authors may want to learn different classes in Omniglot, or learn several different digit datasets (USPS, MNIST, etc) in sequence. See also my comment on stream learning below.

We thank the reviewer for this insightful suggestion. We implemented a new experiment of sequential learning with the datasets USPS and MNIST as suggested, presented in the new Figs. 4(cd), as well as the new Supplementary Note 10. As the MNIST dataset is much larger than the USPS one, we follow the training protocol introduced in (Long et al, Proc. ICCV, p. 2200, 2013) and (Tzeng et al, Proc. of the IEEE CVPR, p. 7167, 2017), where 2,000 training examples are used for MNIST and 1,800 for USPS.

As suggested, the baselines are non-metaplastic networks obtained by partitioning the metaplastic network into two equal parts (each featuring half the number of hidden neurons), and trained independently on each task. We see in the new Fig. 4(c) that the metaplastic network learns sequentially both datasets successfully with accuracies above the baselines, suggesting that for a fixed number of hidden neurons, metaplasticity can provide an increase in capacity.

Due to the topology of convolutional neural network, a network partitioned in two parts has actually less than half the number of parameters as the initial network. For this reason, the new Fig. 4(d) presents a variation of the situation of Fig. 4(c) with the same baselines, and where the metaplastic

network is this time designed with a number of parameters doubled with regards to the baselines (the new Supplementary Note 10 details our methodology). In that case, the accuracy of the metaplastic network matches (but does not exceed) the baselines, despite the fact that the tasks were trained sequentially and not independently as in the baselines.

The idea of a hidden variable is reminiscent also of "Network plasticity as Bayesian inference" Kappel et al, and provides another biologically inspired model of two-level plasticity. The authors may want to compare the current (deterministic) approach with the stochastic approach of Kappel et al.

We have now cited this work in the discussion of our article, as another example of the interest of complex synapses. *"Our proposal, as other proposals of complex synapses with multiple variables⁴¹ or stochastic behaviors⁴², could therefore be an outstanding candidate for nanotechnological implementations"*.

p2157 "penalty term which depends on the previous optimum". One comment here (also correcting a comment in my previous review): While Zenke et al. did use a loss function based on the previous parameter set, their approach does not fundamentally necessitate a separation of tasks. However, they didn't demonstrate it: too bad for them, and good the authors of this manuscript.

Labels for the different curves (network sizes) in Figure 3 are missing.

We have added the labels in the revised version

p81169 "our weight consolidation strategy is tied specifically to the use of a binarized neural network". The implementation of the full precision network is a bit misleading: Performing updates with f_{meta} would effectively induce incorrect gradient in the full precision network. Can a different f_{meta} recover the full precision performance?

We have edited the body text of our article based on this comment. The implementation of the full precision network is now presented more like a control result, and on the other hand we emphasized the qualitative discussion of why we believe that our approach needs the use of an importance parameter (hidden weight for a binarized neural network) in addition to the synaptic weight. We have also replaced *"our weight consolidation strategy is tied specifically to the use of a binarized neural network"* by a more generic *"our weight consolidation strategy is tied specifically to the use of hidden weights"*.

p9l199: The experiments for stream learning are not informative about a key aspect of real-world learning, namely the non-iid sampling of the data. The non-random ordering in stream learning with respect to the categories is a key problem in real-world learning. In the stream learning section, the authors should instead address this, or give a good reason why "All classes are represented in each subset". What happens for instance if all shoes are learned, then all shirts, etc?

In order to address this question, as well as another comment of Reviewer #1, we have now included class incremental training experiments on the CIFAR-10 and CIFAR-100 datasets. These experiments correspond to an extreme case opposite to stream learning: the classes of CIFAR-10 and CIFAR-100 are divided into two subsets. The network is trained on the first classes subset, and then on the second subset, without ever re-seeing examples of the first classes subset.

These new results are described in detail in the new Supplementary Note 11 and the associated Supplementary Figure 8. We have also reported the most important results in the new Figs. 4(efgh).

One other point is that the performance of the proposed solution is highly dependent on how many updates are made, and thus the learning rate. Couldn't the gap between the baseline and the stream learning in fig 5 be reduced by increasing the learning rate?

We have performed an extensive study on the impact of the learning rate on these experiments. We were not able to obtain better results.

p12l301 "For artificial system" -> "For an artificial system"

We thank the reviewer for noticing the typo. It is corrected in the new version.

Reviewer #1 (Remarks to the Author):

Thanks for all the changes.

Reviewer #2 (Remarks to the Author):

Thank you for the detailed responses and clarifications.

Reviewer #3 (Remarks to the Author):

The reviewers have addressed all my comments.

Emre Neftci